# Goedel-Prover-V2: Scaling Formal Theorem Proving with Scaffolded Data Synthesis and Self-Correction

**Yong Lin**[1][*], **Shange Tang**[1][2][*], **Bohan Lyu**[3][*], **Ziran Yang**[1][*], **Jui-Hui Chung**[1][*],
**Haoyu Zhao**[1][*], **Lai Jiang**[7][*], **Yihan Geng**[8][*], **Jiawei Ge**[1], **Jingruo Sun**[4],
**Jiayun Wu**[3], **Jiri Gesi**[6][†], **Ximing Lu**[2], **David Acuna**[2], **Kaiyu Yang**[5][‡],
**Hongzhou Lin**[6][*][†], **Yejin Choi**[2][4], **Danqi Chen**[1], **Sanjeev Arora**[1], **Chi Jin**[1][*]
[1]Princeton Language and Intelligence, Princeton University    [2]NVIDIA
[3]Tsinghua University    [4]Stanford University    [5]Meta FAIR    [6]Amazon
[7]Shanghai Jiao Tong University    [8]Peking University

## Abstract

Automated theorem proving (ATP) — the task of generating a proof that passes automated proof verification given a math question in formal language — is a critical challenge at the intersection of mathematics and Artificial Intelligence (AI). We introduce Goedel-Prover-V2, a family of two language models that establish a new state-of-the-art (SOTA) in open-source ATP, using the Lean proof assistant. In addition to standard expert iteration and reinforcement learning, our approach incorporates three key innovations: (1) During training when improvement plateaus on human questions, the prover does scaffolded data synthesis to generate synthetic questions of increasing difficulty for its own training; (2) The prover is trained to self-correct using Lean compiler feedback; (3) Improved test-time exploration through checkpoint averaging to balance accuracy and diversity. Our small model, Goedel-Prover-V2-8B, reaches 84.6% pass@32 on MiniF2F and outperforms DeepSeek-Prover-V2-671B despite being $80\times$ smaller. Our flagship model, Goedel-Prover-V2-32B, achieves 88.1% on MiniF2F at pass@32 in standard mode and 90.4% in self-correction mode, outperforming prior SOTA by a large margin. Additionally, our flagship model solves 86 problems on PutnamBench at pass@184, securing first place among open-source models and surpassing DeepSeek-Prover-V2-671B's record of 47 problems by pass@1024 with about $20\times$ smaller model size and significantly lower compute budget. Our models, code, and data are released at https://github.com/Goedel-LM/Goedel-Prover-V2.

# 1 Introduction

Automated theorem proving (ATP) is a grand challenge for AI, requiring the construction of step-by-step, machine-verifiable proofs in formal languages like Lean (De Moura et al., 2015; Moura & Ullrich, 2021). The field has advanced rapidly in recent years; for example, AlphaProof (Google DeepMind, 2024), Seed-Prover (Chen et al., 2025) and AlphaGeometry (Trinh et al., 2024; Chervonyi et al., 2025) demonstrated that AI systems are capable of achieving IMO (International Math Olympiad) medal-level performance. While these closed-source frontier models have achieved major leaps in formal theorem proving, they demand extreme-scale compute and remain opaque in their training details. Open-source efforts such as DeepSeek-Prover-V2 (Ren et al., 2025) and Kimina-Prover (Wang et al., 2025) have achieved impressive results on benchmarks like MiniF2F (Zheng et al., 2021) and PutnamBench (Tsoukalas et al., 2024), demonstrating the effectiveness of leveraging the reasoning ability of LLMs through chain-of-thoughts. Yet, there remains substantial room to further improve both the accuracy and efficiency of open models.

---

[*]Core Contributor.

[†]This work is independent of and outside of the work at Amazon.

[‡]All experiments and data processing were conducted outside Meta.

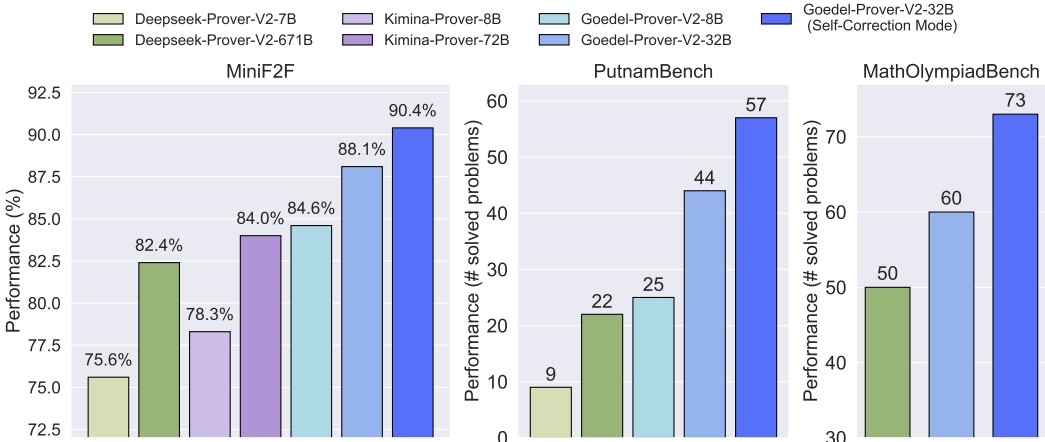

Figure 1: Performance of different models on multiple benchmarks under pass@32.

We aim to push the boundaries of open-source ATP by building stronger and more efficient models. In this work, we introduce Goedel-Prover-V2, a new series of open-source models for automated theorem proving in Lean that establish a new state-of-the-art in both performance and computational efficiency. Given math questions in natural language the verifiers generate whole proofs while leveraging verifier feedback for iterative self-correction. Our flagship 32B model achieves 88.1% pass@32 on the MiniF2F benchmark, improving to 90.4% with self-correction. This performance surpasses both the concurrent 72B Kimina-Prover and the previous SOTA 671B DeepSeek-Prover-V2, while using significantly fewer parameters (Figure 1). On the more challenging PutnamBench, our model solves 44 problems (57 with self-correction), more than doubling the number solved by DeepSeek-Prover-V2 under the same metric. The efficiency of our approach is further underscored by our 8B model, which alone outperforms the 671B DeepSeek-Prover-V2 on MiniF2F despite being nearly 80 times smaller. At the time of this submission, Goedel-Prover-V2 achieves the strongest overall performance among all open-source models.

The key to these gains lies in novel design across the framework, data, and training pipelines. We highlight the main components below:

- **Verifier-guided self-correction:** Even humans need multiple attempts to translate their intuitive proof into Lean. Our incorporate feedback from the Lean compiler (verifier) into the model input to enable the error-correction ability of our theorem prover (*verifier-guided self-correction*). While correcting errors based on verifier feedback has been studied in theorem proving (First et al., 2023) and coding (Olausson et al., 2024; Chen et al., 2024; Bouzenia et al., 2024), we further incorporate this into models that generate long chain-of-thought (CoT) reasoning, which is effective for complex reasoning tasks such as ATP (Jaech et al., 2024; Guo et al., 2025; Wang et al., 2025; Ren et al., 2025).

- **Scaffolded data synthesis:** Successfully combining long CoT with verifier-guided error correction requires special efforts on curating data. In addition to formalizing existing math problems and curating data for error correction, we augment our training statement set through *scaffolded data synthesis*. This technique creates math problems at an appropriate difficulty level to provide the model with better learning signals.

- **Model averaging:** Our training recipe extends beyond standard expert iteration and reinforcement learning. We also apply *model averaging* (Wortsman et al., 2022b) to mitigate the decrease in model output diversity that can occur in the later stages of training.

Our results demonstrate that the frontier of formal theorem proving is attainable within computationally efficient and accessible open-source models. We hope that our open-source theorem prover series, Goedel-Prover-V2, will enable the community to build upon them and accelerate progress toward robust automated solvers that can reliably generate formal proof details, thereby allowing human mathematicians to focus on high-level reasoning while offloading the rigorous verification burden to AI (Kontorovich, 2025).

## 2 METHOD

In this section, we present our method in detail. We start with the key framework innovation compared to the previous vanilla whole-proof generation methods by utilizing the feedback from the Lean compiler to guide the proof-correction procedure (Section 2.1). Then, in Section 2.2, we demonstrate how to curate the training data (statements), with an emphasis on augmenting the data through scaffolded data synthesis. Based on the curated data, we present the details of training our theorem prover in Section 2.3, which includes supervised fine-tuning (SFT), reinforcement learning (RL), and model averaging. Section 2.4 provides a summary of the overall framework.

### 2.1 CHAIN-OF-THOUGHT AND SELF-CORRECTION

Prevailing paradigms for whole-proof generation in ATP have largely been "end-to-end," where a model simply generates a complete formal proof from a theorem statement (Xin et al., 2024b; Lin et al., 2025; Dong & Ma, 2025). Recent work has significantly advanced this approach by leveraging the long-chain-of-thought reasoning capabilities of large models (Wang et al., 2025; Ren et al., 2025).

A key distinction between informal and formal proof generation lies in the availability and utilization of the feedback from formal compilers. Humans naturally leverage such feedback to iteratively revise their proofs. Recent works have shown that integrating the compiler's verification outcomes into proof generation significantly improves synthesis accuracy (First et al., 2023).

Our framework formalizes this intuition by explicitly incorporating verifier feedback within the whole-proof generation loop. We structure the pipeline so that, after an initial proof attempt, verification failures are parsed and communicated back into the model as corrective guidance. The model then generates proof repairs, leading to an iterative self-correction process.

### 2.2 CURATING FORMAL STATEMENTS

In this part, we detail our methods for curating a large, high-quality dataset of Lean 4 statements, which is essential for expert iteration and RL. We start with the formalizer training, which is the core of translating existing math problems written in natural language into Lean statements. We then present our scaffolded data synthesis pipeline that creates math problems at an appropriate difficulty level, which includes a lightweight rule-based method that utilizes the Lean system, and an LLM-based system for large-scale data augmentation.

#### 2.2.1 FORMALIZING STATEMENTS WITH A REASONING FORMALIZER

Existing Lean datasets like Goedel-Pset-v1 (Lin et al., 2025) contain many low-quality problems. Human evaluation on a sampled subset of it shows that over 80% of unsolved problems are incorrectly formalized. This highlights the need for stronger formalizers. We train our formalizer using the standard expert iteration pipeline, which combines Lean syntax checks and semantic evaluation via LLMs to assess translation quality. Only statements that pass both checks are included in the next iteration of SFT.

Table 1: Comparison of different formalizers on 300 Omni-math problems.

| Name | Pass | Failed |
|---|---|---|
| Kimina-autoformalizer | 161 | 139 |
| Goedel-Formalizer-V2 | 228 | 72 |

To initialize expert iteration, we prompt Claude-4-Sonnet to formalize 70K statements with corresponding reasoning traces. Incorporating reasoning capability enables our formalizer to outperform previous models such as Goedel-Formalizer-V1 and Kimina-Autoformalizer, which lack this feature. The total API cost is approximately $1350.

We evaluate our model, Goedel-Formalizer-V2 and Kimina-Autoformalizer on 300 statements from Omni-MATH (Gao et al., 2024a). Results are shown in Table 1. To validate the reliability of the LLM judge, we performed a human evaluation on 30 problems, which showed 92% agreement between the LLM and human expert. The training and evaluation details are provided in Appendix D.

We use Goedel-Formalizer-V2 to formalize 300K informal statements from the OpenMathReasoning (OMR) dataset (Moshkov et al., 2025) and scaffolded informal statements (see Section 2.2.2).

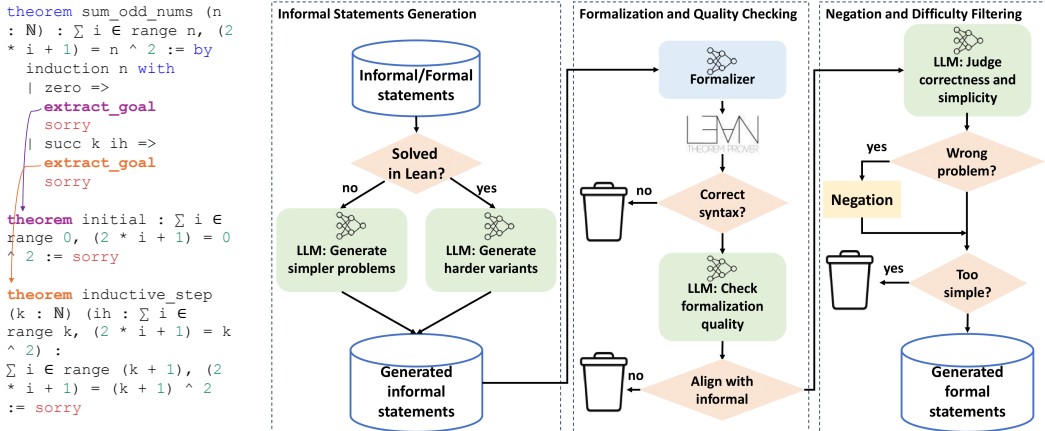

Figure 2: **(Left)** Scaffolding from formal proofs with `extract_goal`. **(Right)** Scaffolding from informal problems, which leverages LLMs to generate new informal problems, formalize them, and filter them for difficulty and correctness.

### 2.2.2 SCAFFOLDED DATA SYNTHESIS

As shown in Figure 2, our scaffolded data synthesis pipeline includes scaffolding from both formal proofs and informal problems.

**Rule-based scaffolding.** When the prover fails to prove a theorem, we can still leverage the incomplete proof attempt to generate simpler, related problems. The intuition is that even an incorrect proof may contain valid subgoals that represent easier lemmas. We use a powerful tactic in Lean, `extract_goal`, to capture these unsolved states from a proof attempt. These extracted goals, which are well-formed mathematical statements, are then used to augment our training data. Since an extracted statement is not guaranteed to be provable, we also include its negation, thereby training the model to recognize both true and false propositions.

**LLM-based scaffolding.** Another data synthesis approach is to leverage an LLM's strong natural language reasoning ability to create new problems. For a given theorem, we prompt an LLM (Qwen3-32B) to generate simpler sub-problems if the theorem is unsolved, or harder variants if it is already solved. The generated informal statements are then formalized using our formalization pipeline with Goedel-Formalizer-V2. To avoid the high inference cost of verifying incorrect or trivial statements, we use an LLM-based filter to assess each statement for correctness and difficulty. Trivial statements are discarded, while the negations of incorrect statements are added to the dataset. This filtering process significantly accelerates data synthesis with only a minor trade-off in potentially discarding some valid problems due to LLM judgment errors. Further details are in Appendix E.

### 2.3 TRAINING ALGORITHMS

**Supervised fine-tuning and expert iteration.** We follow the standard pipeline of Expert Iteration by iterating between using the model to conduct large-scale inference on the statement sets, collecting correct proofs with reasoning traces, and fine-tuning the model further on the collected samples. We utilize the open-source framework Llama-Factory (Zheng et al., 2024) for model training.

**Reinforcement learning.** We aim to train a model that can both generate complete proofs and correct its own errors using verifier-guided feedback. To achieve this efficiently, our RL implementation adopts a multi-task setup: 50% of the inputs for whole proof generation, and the other 50% for self-correction. We train for a single epoch, using 46K and 67K unique inputs for the 8B and 32B models, respectively. On the algorithmic side, we use a hybrid GRPO-based approach (Shao et al., 2024): we remove group normalization as suggested by Dr.GRPO (Liu et al., 2025) to avoid inherent bias on length, incorporate clip-higher, overlong penalties, and dynamic sampling from DAPO (Yu et al., 2025), and exclude the KL regularization term from the objective to encourage exploration. A key observation is that question difficulty significantly impacts RL training. To address this, we

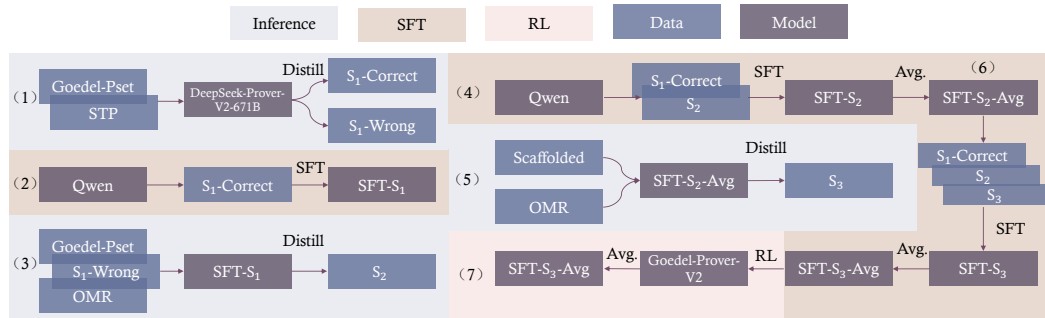

Figure 3: The overall pipeline of model training.

modify the dynamic sampling strategy to only include problems with pass rates in the range (0, 0.75] during optimization. For further implementation details, see Appendix F.1.

**Model averaging for enhanced diversity.** We observed that in the later stages of SFT and RL, the model's diversity decreases. This is reflected by an increase in pass@1 but a decline in pass@N for larger values of N like 32. We adopt model averaging to alleviate this issue and enhance model diversity (Wortsman et al., 2022b;a; Lin et al., 2023a;b; Dang et al., 2025). Specifically, let the parameters of the base model be denoted as $\theta_0$, and those of the fine-tuned model as $\theta$. We use the combined model parameters defined as $(1 - \alpha)\theta_0 + \alpha\theta$, where $\alpha \in (0, 1)$. Existing literature has demonstrated that this simple approach can significantly improve the feature diversity of the final model. Our observations also confirm that this method effectively enhances pass@N. We perform model averaging at each stage of the process. Specifically, we apply model averaging after each iteration of SFT and after RL. We use $\alpha = 0.8$ in our main experiments.

## 2.4 WHOLE PIPELINE: PUTTING EVERYTHING TOGETHER

We observe that while DeepSeek-Prover-V2 models are already heavily trained and have lost self-correction capabilities, other models like Qwen3 lack the ability to generate formal proofs. To address this trade-off, we use data distilled from DeepSeek-Prover-V2 to cold-start Qwen3, followed by large-scale generation of revision and direct proof data with the resulting model. We then train our own model and iteratively refine it, incorporating scaffolded data. During training, we observe a reduction in output diversity (a form of overfitting) after each stage and apply model averaging to mitigate this. The whole training pipeline consists of the following steps, as illustrated in Figure 3:

(1) We distill DeepSeek-Prover-V2 models on Goedel-Pset and STP (Dong & Ma, 2025), creating the cold-start dataset $S_1$, which includes correct proofs ($S_1$-Correct) and incorrect ones ($S_1$-False).

(2) We fine-tune Qwen3 with $S_1$-Correct to obtain the model SFT-$S_1$.

(3) Using SFT-$S_1$, we perform proof revision on $S_1$-False and whole-proof generation on both Goedel-Pset and OMR. We collect the correct proofs to form dataset $S_2$.

(4) We fine-tune Qwen3 with $930k$ entries from $S_1$-Correct and $S_2$ to create SFT-$S_2$. Subsequently, we perform model averaging between SFT-$S_2$ and Qwen3 to obtain SFT-$S_2$-Avg.

(5) We use SFT-$S_2$-Avg to generate proofs on our synthesized scaffolded statements and OMR. The correct proofs are collected to form dataset $S_3$.

(6) We fine-tune SFT-$S_2$-Avg $860k$ entries from $S_1$-Correct, $S_2$, and $S_3$ to yield SFT-$S_3$. We then perform model averaging between SFT-$S_3$ and Qwen3 to produce SFT-$S_3$-Avg.

(7) Finally, we train SFT-$S_3$-Avg with RL, apply model average, and obtain the final models.

**Compute.** We conduct SFT and RL training of the 32B model on 64 GPUs, and the 8B model on 32 GPUs. The total GPU hours for dataset generation is approximately 12k H100 GPU hours. For the 8B model, SFT and RL used 2.3k and 1.3k GPU hours, respectively; the 32B model required 9.2k GPU hours for SFT and 3.9k GPU hours for RL.

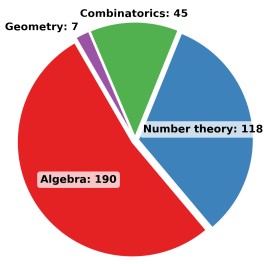

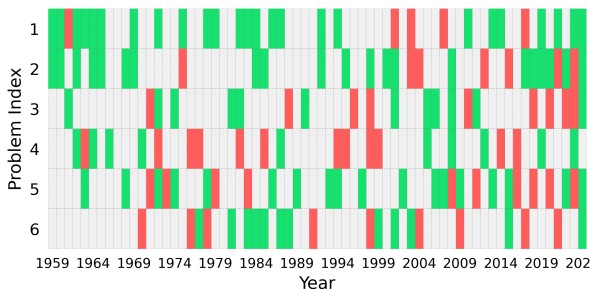

Figure 4: Distribution of problems in MathOlympiadBench by category.

Figure 5: Community's achievement on IMO problems by year and problem index. Each cell represents a problem: ⬜ statement not formalized, 🟥 not solved, and 🟩 solved.

**Decontamination.** Comprehensive measures ensure separation between our training data and evaluation benchmarks. First, the source of informal statements, like OMR, has already undergone rigorous decontamination against common mathematical benchmarks. Since formal benchmarks like miniF2F are largely derived from these same informal benchmarks, e.g., MATH, the decontamination of the source informal data provides a strong initial safeguard. Second, following standard practices in recent literature (Dong & Ma, 2025), we compare the formal statements in the training set and the test set to ensure there's no training sample that has an exact string match with the test instances.

## 3 EXPERIMENTS

This section presents our experiment results on Goedel-Prover-V2. We first discuss our evaluation benchmarks (Section 3.1). Then, we discuss our main evaluation results on the selected benchmarks (Section 3.3), and correspondingly the scaling behavior (Section 3.5). Finally, we investigate the performance of reinforcement learning and model averaging (Section 3.6).

### 3.1 BENCHMARKS

**MiniF2F.** MiniF2F (Zheng et al., 2021) consists of 488 problem statements (244 validation and 244 test problems) in Lean. The problems are drawn from high-school level competitions including the AMC, AIME, and the International Mathematical Olympiad (IMO). We use the version of MiniF2F provided by Kimina (Wang et al., 2025)[1], which has some incorrect statements fixed.

**PutnamBench.** PutnamBench (Tsoukalas et al., 2024) focuses on college-level mathematics competition problems that are sourced from the William Lowell Putnam Mathematical Competition years 1962 - 2023. PutnamBench comprises 644 problems, covering algebra, analysis, number theory, geometry, combinatorics, probability, and set theory.

**MathOlympiadBench.** To facilitate open evaluation on the most challenging math competition problems, we constructed MathOlympiadBench. It comprises 360 formalized IMO and IMO short lists problems, sourced from Compfiles [2] and IMOSLLean4 repository [3]. It contains 158 IMO problems from 1959 to 2024, 131 IMO shortlist problems covering 2006 to 2023, 68 national mathematical Olympiad problems, and 3 additional mathematical puzzles. The statistic of problem categories in MathOlympiadBench is presented in Figure 4, and Figure 5 visualizes humans' formalizing and solving status of IMO problems in MathOlympiadBench. Appendix A provides more details of MathOlympiadBench and its comparison with MiniF2F.

We also evaluate models on FIMO (Liu et al., 2023), which comprises problems from IMO Shortlisted problems from 2006 to 2021, and DeepSeek-ProverBench (Ren et al., 2025) which includes both high-school competition problems and undergraduate-level problems.

---

[1] https://huggingface.co/datasets/AI-MO/minif2f_test
[2] https://dwrensha.github.io/compfiles/imo.html
[3] https://github.com/mortarsanjaya/IMOSLLean4

## 3.2 EVALUATED METHODS

**Lean version.** The evaluations are done under Lean 4.9.0-rc1 for fair comparison with previous works (Xin et al., 2024b; Lin et al., 2025; Dong & Ma, 2025; Ren et al., 2025).

**Token budgets.** For the first round of whole-proof generation, the max token length of the model is set to be 30,000. For the verifier-guided error-correction, we sequentially conduct 2 additional rounds of self-correction, given the verifier's feedback on the previous attempt, where the total number of tokens is set to be 40,000. We report the pass@N metric.

## 3.3 MAIN RESULTS

The evaluation results on MiniF2F and Putnam-Bench are shown in Table 2 and Table 3 respectively, the results on FIMO and DeepSeek-ProverBench are shown in Table 4. The results for MathOlympiadBench are presented in the rightmost figure of Figure 1. Below, we summarize and discuss the results.

**High performance at modest scale.** Our 32B model achieves pass@32 accuracy of 88.1% on MiniF2F, with 90.4% after 2 rounds of error-correction, exceeding the previous state-of-the-art DeepSeek-Prover-V2-671B's 82.4% while using far fewer parameters. Even the 8B variant

Table 2: Performance of different whole-proof generation methods on MiniF2F test split. [†] denotes concurrent work

| Method | Budget | Performance |
|---|---|---|
| Goedel-Prover-SFT | 32 | 57.6% ± 0.7% |
| (Lin et al., 2025) | 3200 | 62.7% |
| STP (Dong & Ma, 2025) | 128 | 61.2% ± 0.6% |
| | 3200 | 65.0% ± 0.5% |
| | 25600 | 67.6% |
| Kimina-Prover-72B | 32 | 68.85% |
| (Wang et al., 2025) | 8192 | 80.74% |
| DeepSeek-Prover-V2-7B | 32 | 75.6% ± 0.5% |
| (Ren et al., 2025) | 8192 | 82.0% |
| DeepSeek-Prover-V2-671B | 32 | 82.4% ± 0.6% |
| | 8192 | 88.9% |
| Kimina-Prover-8B[†] | 32 | 78.3% |
| Kimina-Prover-70B[†] | 32 | 84.0% |
| (Wang et al., 2025) | 1024 | 87.7% |
| w/ TTRL | unknown | 92.2% |
| Goedel-Prover-V2-8B | 32 | 84.6% ± 0.6% |
| | 1024 | 87.9% |
| | 8192 | 90.2% |
| w/ self-correction | 32 | 86.7% ± 0.2% |
| | 1024 | 89.3% |
| Goedel-Prover-V2-32B | 32 | 88.1% ± 0.8% |
| | 1024 | 91.8% |
| | 8192 | 92.2% |
| w/ self-correction | 32 | 90.4% ± 0.6% |
| | 1024 | 92.6% |

achieves 84.6%, nearly matching or outperforming Kimina-Prover-70B's results under the same inference budget, and outperforming the previous SOTA DeepSeek-Prover-V2-671B on MiniF2F, with a significantly smaller model size. On PutnamBench, our 32B model solves 43 problems under pass@32, nearly doubling the DeepSeek-Prover-V2-671B model's performance under the same budget. With error-correction, Goedel-Prover-V2-32B solves 57 problems under pass@32, outperforming DeepSeek-Prover-V2-671B under pass@1024. Under pass@184 and error correction, Goedel-Prover-V2-32B solves 86 on PutnamBench, securing the best open-source theorem prover on the leaderboard.

Table 3: Comparison of different models on PutnamBench. Our Goedel-Prover-V2 with compiler-guided self-correction solves 86 problems, improving the previous SOTA (DeepSeek-Prover-V2) by 39 more problems, and securing the best open-source model on the leaderboard.

| Model | num-solved | open-source | compute |
|---|---|---|---|
| **Goedel-Prover-V2 (self-correction mode)** | **86** | ✓ | pass@184 |
| **Goedel-Prover-V2 (self-correction mode)** | **57** | ✓ | pass@32 |
| **Goedel-Prover-V2** | **43** | ✓ | pass@32 |
| DeepSeek-Prover-V2 | 47 | ✓ | pass@1024 |
| DeepSeek-Prover-V2 | 22 | ✓ | pass@32 |
| DSP+ | 23 | ✓ | pass@128 |
| Kimina-Prover-7B-Distill | 10 | ✓ | pass@192 |
| Self-play Theorem Prover | 8 | ✓ | pass@3200 |
| Goedel-Prover-SFT | 7 | ✓ | pass@512 |
| SeedProver | 331 | ✗ | unknown |
| ABEL | 7 | ✗ | pass@596 |

Table 4: Experimental Results on FIMO and DeepSeek-ProverBench. Model names are abbreviated, where DS is for DeepSeek, Goedel is for Goedel-Prover-V2, and rev. is for w/ revision, etc.

| Dataset | DS-V2-7B | Kimina-8B | Kimina-72B | Goedel-8B | Goedel-8B rev. | Goedel-32B | Goedel-32B rev. |
|---|---|---|---|---|---|---|---|
| **FIMO** | 5.70 | 3.35 | 5.70 | 7.05 | 7.05 | 7.05 | **9.06** |
| **ProverBench** | 0.31 | 1.38 | 1.53 | 1.53 | 1.53 | 1.85 | **2.46** |

**Efficacy of verifier-guided self-correction.** Adding self-correction provides a consistent gain of approximately 2 percentage points in pass@32 for our models on MiniF2F. On PutnamBench, error correction leads to 14 more solves under pass@32. Moreover, DeepSeek-Prover-V2-7B, which was not explicitly trained with revision, shows negligible improvement (from 75.8% to 76.2%) on MiniF2F with under the self-correction mode. This shows that the self-correction method is primarily effective when the LLM is trained to interpret and utilize Lean compiler feedback.

**Sample-efficient inference.** Unlike prior models such as Kimina-Prover and DeepSeek-Prover-V2, which rely heavily on large sampling budgets or test-time reinforcement learning to reach peak accuracy, Goedel-Prover-V2 attains very high pass@N with minimal inference overhead (N=32 or 64), indicating that the model internalizes powerful reasoning strategies during training. The sample efficiency, together with the relatively small size, makes Goedel-Prover-V2 series a very good candidate for the community to develop new algorithms and test on different benchmarks.

## 3.4 ANALYSIS OF SELF-CORRECTION

In our main experiments, self-correction was conducted for a maximum of 2 rounds with a 40k token context length. To further explore its capabilities, we used YaRN (Peng et al., 2024) to extend the context length to 128k tokens and allowed up to 5 revision iterations. We conducted a series of experiments on the MiniF2F benchmark at pass@32, the results of which are presented in Figure 6. Alongside our standard prompting setup (Self Correction), we performed two ablation studies: (1) removing the specific compiler error messages (w/o Error Messages), and (2) removing the chain-of-thought from previous attempts, retaining only the formal proof (w/o Previous CoTs).

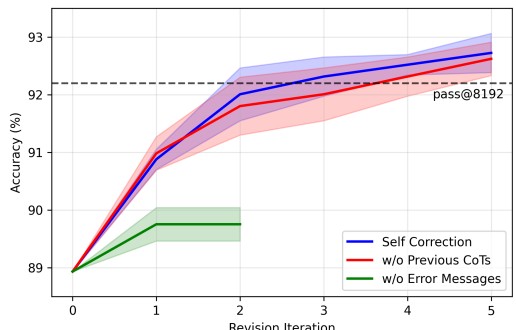

Figure 6: Ablation study on self-correction with extended context length and revision iterations on the MiniF2F test split at pass@32.

The results show that removing compiler feedback significantly lowers performance, confirming that specific error messages are crucial for effective revision. In contrast, removing the reasoning from previous attempts yields very minimal performance difference, which suggests that including prior CoT does not substantially improve performance. Since including CoT incurs additional compute cost per task, it may be preferable to omit this information for greater efficiency. Furthermore, with an extended context and more revision iterations, the full self-correction model's pass@32 accuracy on MiniF2F reaches an average of 92.7%, which surpasses the 92.2% performance of the model without self-correction at pass@8192, highlighting the efficiency of our iterative revision process.

## 3.5 SCALING ANALYSIS

Figure 7(a) illustrates the scaling behavior of Goedel-Prover-V2 (8B and 32B variants) across different inference budgets compared against baselines. At the lower sampling regime (pass@32), Goedel-Prover-V2-32B already achieves 88.1%, notably surpassing DeepSeek-Prover-V2-671B (82.4%) and Kimina-Prover-72B (84.0%). This advantage persists across inference budgets, with our self-correction mode providing approximately a 2-point performance improvement under pass@32 and pass@64, peaking at 92.6% at pass@8192. The smaller 8B model also demonstrates strong scalability, surpassing the 671B DeepSeek model at all budgets.

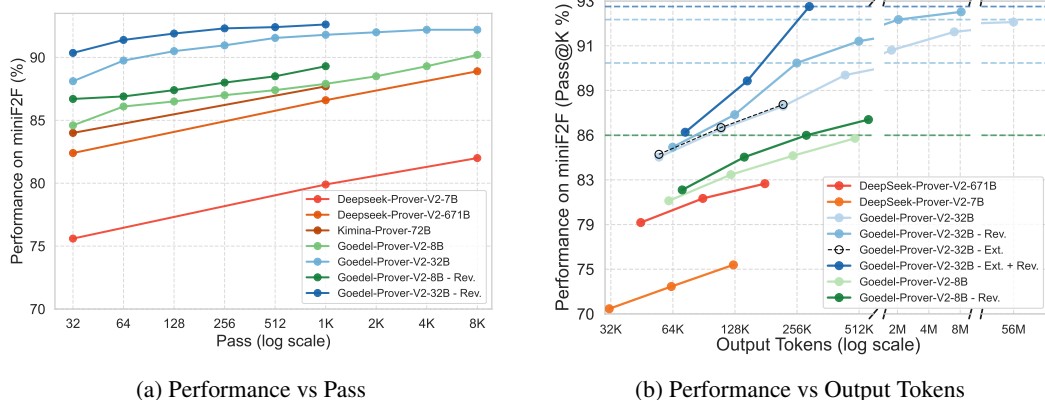

(a) Performance vs Pass

(b) Performance vs Output Tokens

Figure 7: Scaling behavior on MiniF2F test. 'Rev.' denotes 2 rounds of self-correction within 40k context length, 'Ext.' extending the context length to 128k but allowing only the first round generation, and 'Ext. + Rev.' demotes 5 rounds of self-correction within 128k context length.

To further analyze efficiency from a compute perspective, we consider FLOPs with a commonly used estimate of model size × output length (Snell et al., 2024). In Figure 7(b), we also incorporate the setting introduced in Section 3.4, extending the context length to 128k tokens and allowing up to 5 revision iterations ('Ext. + Rev.'). Meanwhile, we also add a baseline that only extends the context length but does not allow more rounds of revision ('Ext.'). The results demonstrate that, for the same volume of output tokens, Goedel-Prover-V2 outperforms all baselines, with self-correction yielding further efficiency gains. Notably, while two rounds of revision increase the average output length by less than 20%, the model achieves performance on par with doubling the number of passes in the vanilla setting. Furthermore, when extending the context window and allowing more rounds of self-correction, the model at pass@32 surpasses the performance of the vanilla setting at pass@8192, despite utilizing only $1/218$ of the compute. Comparing 'Ext. + Rev.' and 'Ext.', we can also see that simply extending the context length does not help; the improvement comes from revision. This indicates that scaling revision is more effective than scaling passes in the vanilla setting.

We provide detailed data of model performance and average output lengths in Appendix B.

## 3.6 RL AND MODEL AVERAGING

We systematically evaluate the impact of RL steps and model averaging strategies applied to them on the performance of the final models. Specifically, for RL checkpoints at steps 60, 80, and 90, we apply model averaging with coefficients $\alpha = 0.6, 0.7, 0.8, 0.9$ (where $\alpha$ is the weight of the base model). We assess each averaged model in both vanilla (whole-proof generation) and correction (with self-correction) settings, evaluating both pass@1 and pass@N, as visualized in Figure 8.

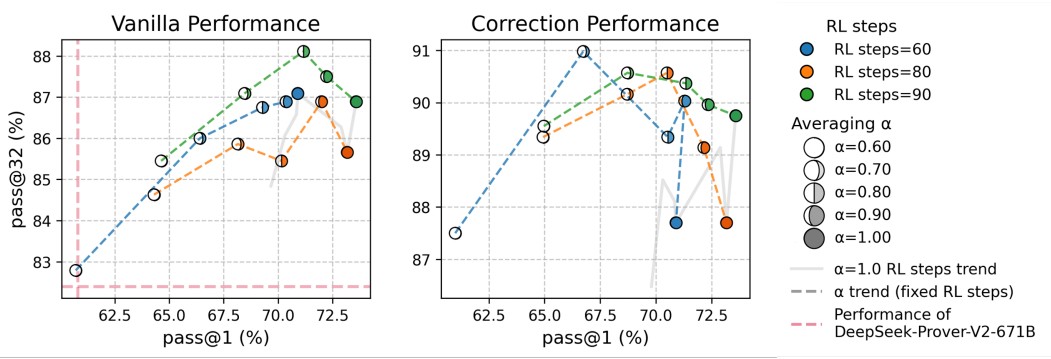

Figure 8: The effects of varying RL steps and model averaging ratios on the pass@1 and pass@32 performance, both with and without correction, along with the DeepSeek-Prover-V2-671B baseline.

For both vanilla and correction settings, pass@1 consistently increases as the number of RL steps grows. For vanilla pass@N, performance stabilizes at higher RL steps, whereas in the correction setting, pass@N continues to improve. This indicates that correction benefits more from RL, likely due to the shortage of high-quality self-correction data in the SFT stages.

Examining different values of $\alpha$, we observe that a higher proportion of the base model (i.e., lower $\alpha$) leads to lower pass@1, but pass@N first rises and then falls as $\alpha$ increases. There exists an optimal model averaging ratio that maximizes pass@N, which holds for both vanilla and correction settings.

## 4    RELATED WORKS

Recent advancements in ATP with LLMs have explored several distinct strategies, apart from raw end-to-end generation (Xin et al., 2024a; Lin et al., 2025; Wang et al., 2025), where a model simply generates a complete formal proof from a theorem statement.

**Proof search methods.** To improve correctness over single-pass generation, many systems employ proof search to incrementally construct proofs with verifier feedback at each step (Wu et al., 2024; Li et al., 2024; Xin et al., 2024b; 2025). Hybrid approaches combine a powerful LLM for high-level sketching with dedicated provers for detailed steps (Cao et al., 2025; Zhou et al., 2025). Similarly, Seed-Prover (Chen et al., 2025) relies on extensive test-time search and refinement, demanding substantial computational resources.

**Verifier-Guided refinement and self-repair.** A growing body of work focuses on verifier-guided refinement. Some methods use auxiliary signals, such as informal sketches (Jiang et al., 2023; Gloeckle et al., 2024; Cao et al., 2025) or retrieved theorems (Yang et al., 2024), to guide generation. Another trend enables models to iteratively revise candidate proofs based on feedback Ji et al. (2025); Baba et al. (2025); Zhou et al. (2025); Ren et al. (2025); Wang et al. (2025); Chen et al. (2025), improving success rates more efficiently than exhaustive search. Goedel-Prover-V2 builds directly on this trajectory by adopting a self-revision framework. It adapts general self-repair paradigms from coding and reasoning (Yao et al., 2023; Shinn et al., 2023; First et al., 2023; Olausson et al., 2024; Chen et al., 2024; Bouzenia et al., 2024) to the domain of formal mathematics, enabling the model to propose and refine proofs iteratively until they are formally verified.

## 5    CONCLUSIONS AND FUTURE DIRECTIONS

Even though Goedel-Prover-V2 advances the SOTA in open-source performance, several promising directions remain for future improvement.

**Reasoning versus knowledge retrieval.** Our evaluation on ProofNet reveals a distinction between cognitive reasoning and library-dependent knowledge (see Appendix G). While our model excels on reasoning-intensive benchmarks, it faces challenges on ProofNet, which emphasizes the utilization of abstract tactics from Mathlib. Our experiments show that while Retrieval-Augmented Generation (RAG) significantly boosts performance on ProofNet, it does not improve (and may even hinder) performance on MiniF2F. This suggests that future models must balance intrinsic reasoning with the ability to dynamically retrieve and apply complex library-specific tactics, addressing the trade-off between self-contained problem solving and knowledge-intensive formalization.

**Generalization across domains and environments**. Regarding the domain, current works are influenced by benchmarks like MiniF2F that skew towards algebra and number theory. This introduces a topic bias that hinders generalization to underrepresented fields like geometry, combinatorics, and advanced analysis. Integrating rich supervision from textbooks and journals would be beneficial. Regarding the environment, our models are standardized on Lean 4.9. Generalizing to newer Lean and Mathlib versions would allow the model to leverage more powerful tactics and optimizations, though this requires addressing the challenge of continuously adapting the models to new tactics.

**Metacognition and self-improvement**. Recent work on LLM metacognition (Didolkar et al., 2024) suggests that models can benefit from explicitly reflecting on their reasoning process and augmenting the missing skills. Applying these ideas to theorem proving could allow Goedel-Prover-V2 to better assess when its reasoning is on track, when to generate subgoals, or when to seek additional information—potentially reducing reliance on brute-force sampling.

ACKNOWLEDGEMENT

This project was the result of a close collaborative effort by all authors, and every major component benefited from joint discussion and iteration. The author would like to thank Igor Gitman for support with the curation of self-correction data.

HZ and SA acknowledge the support from Schmidt, Darpa, ONR, and NSF. CJ acknowledges the support from NSF-OAC-2411299, NSF-IIS-2239297, and Princeton AI Lab Seed Fund.

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

APPENDIX

# A  DETAILS OF MATHOLYMPIADBENCH

**Construction.**  MathOlympiadBench is human-processed to eliminate several issues presented in the source problems: 1. incomplete problem statements, 2. distribution across multiple files, 3. multiple theorems per problem, and 4. incompatibility with the commonly used Mathlib. The verification process ensures that each problem contains exactly one formal theorem with its corresponding informal statement, and confirms that all formal statements can pass the compilation with the `sorry` tactic.

**Comparison with MiniF2F.**  There are 20 IMO problems in the test set of MiniF2F. We manually examined all problems shared between MathOlympiadBench and MiniF2F and identified 3 cases in MiniF2F exhibiting issues such as: 1. the formal statement to be proved is strictly weaker than the informal statement, and 2. the formal statement does not match the informal statement.

In the following, we present these 3 cases comparing the problem formalizations in MiniF2F and MathOlympiadBench.

## A.1  IMO 1981, PROBLEM 6: SPECIFIC VALUE VS. GENERAL FORMULA

**MathOlympiadBench**

```
1  /-!
2  # International Mathematical
       Olympiad 1981, Problem 6
3
4  Suppose that f : ℕ × ℕ → ℕ
       satisfies
5
6   1) f (0, y) = y + 1
7   2) f (x + 1, 0) = f (x, 1),
8   3) f (x + 1, y + 1) = f (x, f (x
       + 1, y))
9
10 for all x y ∈ ℕ.
11
12 Determine f (4, 1981).
13 -/
14
15 def no_eval (x : ℕ) : ℕ := x
16 abbrev solution_value : ℕ :=
17   no_eval ((2^·)^[1984] 1 - 3)
18
19 theorem imo1981_p6 (f : ℕ → ℕ →
       ℕ)
20   (h1 : ∀ y, f 0 y = y + 1)
21   (h2 : ∀ x, f (x + 1) 0 = f x 1)
22   (h3 : ∀ x y, f (x + 1) (y + 1) =
23       f x (f (x + 1) y)) :
24   f 4 1981 = solution_value :=
       sorry
```

**MiniF2F**

```
1  /--
2  The function f(x,y) satisfies
3
4  (1)  f(0,y) = y + 1,
5
6  (2)  f(x + 1, 0) = f(x, 1),
7
8  (3)  f(x + 1, y + 1) = f(x, f(x + 1, y)),
9
10 for all non-negative integers x, y
       . Determine f(4, 1981).
11 -/
12
13 theorem imo_1981_p6 (f : ℕ → ℕ →
       ℕ)
14   (h₀ : ∀ y, f 0 y = y + 1)
15   (h₁ : ∀ x, f (x + 1) 0 = f x 1)
16   (h₂ : ∀ x y, f (x + 1) (y + 1) =
17       f x (f (x + 1) y)) :
18   ∀ y, f 4 (y + 1) = 2 ^ (f 4 y +
       3) - 3 := by
```

The informal statement requires computing the exact value of $f(4, 1981)$, a specific number. However, the MiniF2F formalization only asks to prove a general recurrence relation, which is an intermediate step in the solution process. This discrepancy can make the formal proof substantially easier than solving the original problem. In contrast, MathOlympiadBench faithfully translates the original problem into formal language, requiring the proof of the final numerical value as demanded by the IMO statement.

## A.2 IMO 1983, PROBLEM 6: INCOMPLETE VS. FULL CONDITION

**MathOlympiadBench**

```
1  /-!
2  # International Mathematical
       Olympiad 1983, Problem 6
3
4  Suppose that a,b,c are the side
       lengths of a triangle. Prove
       that
5
6     a²b(a − b) + b²c(b − c) + c²a(c − a) ≥ 0.
7
8  Determine when equality occurs.
9  -/
10
11 abbrev EqualityCondition (a b c :
       ℝ) : Prop :=
12   a = b ∧ a = c
13
14 theorem imo1983_p6
15   (T : Affine.Triangle ℝ
       (EuclideanSpace ℝ (Fin 2))) :
16   let a := dist (T.points 1)
       (T.points 2)
17   let b := dist (T.points 0)
       (T.points 2)
18   let c := dist (T.points 0)
       (T.points 1)
19   0 ≤ a^2*b*(a-b) + b^2*c*(b-c) +
       c^2*a*(c-a) ∧
20   (0 = a^2*b*(a-b) + b^2*c*(b-c) +
       c^2*a*(c-a) ↔
21   EqualityCondition a b c) :=
       sorry
```

**MiniF2F**

```
1  /-- Let a, b and c be the lengths
       of the sides of a triangle.
       Prove that
2
3  a²b(a − b) + b²c(b − c) + c²a(c − a) ≥ 0.
4
5  Determine when equality occurs.
6  -/
7
8  theorem imo_1983_p6 (a b c : ℝ)
9    (h₀ : 0 < a ∧ 0 < b ∧ 0 < c)
10   (h₁ : c < a + b) (h₂ : b < a +
       c)
11   (h₃ : a < b + c) :
12   0 ≤ a^2*b*(a-b) + b^2*c*(b-c) +
       c^2*a*(c-a) := by
```

The informal statement has two parts: proving an inequality and determining the condition for equality. The MiniF2F version only formalizes the inequality, completely omitting the second part of the problem. In contrast, MathOlympiadBench provides a complete formalization.

## A.3 IMO 1962, PROBLEM 2: INFORMAL & FORMAL STATEMENT MISMATCH

**MiniF2F**

```
1 /-- Determine all real numbers x
      which satisfy the inequality:
2
3 √(√(3 - x) - √(x + 1)) > 1/2
4
5 Show that it is [-1, 1 - √127/32).
6 -/
7
8 theorem imo_1962_p2 (x : ℝ) (h₀ :
      0 ≤ 3 - x) (h₁ : 0 ≤ x + 1)
9    (h₂ : 1 / 2 < Real.sqrt (3 -
      x) - Real.sqrt (x + 1)) : -1 ≤
      x ∧ x < 1 - Real.sqrt 31 / 8
      := sorry
```

**MathOlympiadBench**

```
1  /-!
2  # International Mathematical
      Olympiad 1962, Problem 2
3
4  Determine all real numbers x
      which satisfy
5
6  √(3 - x) - √(x + 1) > 1/2.
7  -/
8
9  abbrev SolutionSet : Set ℝ :=
      Set.Ico (-1) (1 - √31 / 8)
10
11 theorem imo1962_p2 (x : ℝ) :
12   x ∈ SolutionSet ↔
13   x ≤ 3 ∧ -1 ≤ x ∧ 1/2 < √(3 - x)
      - √(x + 1) := sorry
```

For this problem, the original inequality appears in two different versions[4]: one as $\sqrt{3 - x} - \sqrt{x + 1} > \frac{1}{2}$ and another as $\sqrt{\sqrt{3 - x} - \sqrt{x + 1}} > \frac{1}{2}$. In MiniF2F, the informal statement uses the latter (the nested square root version), but the formal statement is based on the former (the simpler difference of square roots), resulting in a mismatch between the informal and formal statements. In contrast, MathOlympiadBench ensures that both the informal and formal statements consistently correspond to the same version of the problem.

## B SCALING BEHAVIOR

Table 5 reports the performance (%) of Goedel-Prover-V2 (8B and 32B) on MiniF2F under varying inference budgets, both in vanilla and self-correction modes. As shown, self-correction consistently improves accuracy at all compute levels, and both model sizes demonstrate strong scaling behavior.

| Model | 32 | 64 | 128 | 256 | 512 | 1024 | 2048 | 4096 | 8192 |
|---|---|---|---|---|---|---|---|---|---|
| 32B (self-correction mode) | 90.4 | 91.4 | 91.9 | 92.3 | 92.4 | 92.6 | – | – | – |
| 32B | 88.1 | 89.8 | 90.5 | 91.0 | 91.6 | 91.8 | 92.0 | 92.2 | 92.2 |
| 8B (self-correction mode) | 86.7 | 86.9 | 87.4 | 88.0 | 88.5 | 89.3 | – | – | – |
| 8B | 84.6 | 86.1 | 86.5 | 87.0 | 87.4 | 87.9 | 88.5 | 89.3 | 90.2 |

Table 5: The performance (%) of Goedel-Prover-V2 on MiniF2F across different compute budget.

Table 6 shows the average number of output tokens per sample for each model and revision setting. These statistics allow for a more precise estimate of computational cost, since FLOPs are commonly estimated as model size multiplied by output length. Notably, even with two rounds of revision, the increase in average output length is less than 20% compared to the vanilla setting, showing that self-correction provides efficient use of compute.

| Model | Vanilla | Revision 1 | Revision 2 |
|---|---|---|---|
| DeepSeek-Prover-V2-7B | 3930.8 | – | – |
| DeepSeek-Prover-V2-671B | 5588.3 | – | – |
| Goedel-Prover-V2-8B | 7641.9 | 8551.0 | 8878.0 |
| Goedel-Prover-V2-32B | 6847.7 | 7587.2 | 7978.2 |

Table 6: Average tokens per sample across different models and settings.

---

[4]Please refer to: https://artofproblemsolving.com/wiki/index.php/1962_IMO_Problems/Problem_2.

As shown, revision-based self-correction achieves better performance with only a moderate increase in output length, highlighting its computational efficiency.

Overall, these results indicate that Goedel-Prover-V2 efficiently internalizes reasoning during training, requiring fewer inference samples and output tokens to achieve comparable or superior accuracy. The consistent gains from verifier-guided self-correction across all inference budgets underscore the value of combining error correction with long-chain-of-thought reasoning in formal theorem proving.

## C  FORMAL NEGATION

We attempt to disprove unsolved statements by proving their logical negation. This is achieved by parsing the Lean 4 statements as follows.

```
theorem fourIsPrime (a : ℕ) (ha : a = 4) : a.Prime := by sorry
theorem fourIsPrimeNeg : ¬ ∀ (a : ℕ) (ha : a = 4), a.Prime := by sorry
```

## D  DETAILS FOR TRAINING FORMALIZER

We initialize the pipeline by fine-tuning Qwen3-32B on 70K statements distilled from OMR using Claude-4-Sonnet, yielding Goedel-Formalizer-iter0. This model formalizes 300K OMR statements, which are filtered through syntax and semantic checks to create the training set for Goedel-Formalizer-iter1. Repeating this process results in our final model, Goedel-Formalizer-V2.

For semantic verification, we employ Qwen3-32B as a judge. We enforce a strict consensus mechanism: a formalization is considered faithful only if the judge approves it across four independent queries.

To validate this judge, we manually evaluated a sample of 30 problems. After excluding 5 syntactically incorrect instances, we analyzed the remaining 25. While the LLM correctly rejected 3 inappropriate formalizations, it failed to detect 2 subtle errors identified by humans. Overall, this yields a 92% (23/25) agreement rate between the LLM and human annotators, confirming the judge's reliability.

Here is the exact prompt for LLM judging the faithfulness of formalization.

```
You will receive a math problem consisting of its natural language
    statement along with its formal statement in LEAN 4.

Please evaluate whether the formal LEAN statement appropriately
    translates the natural language statement based on the following
    criteria:

1. Key Elements: The problem's essential components are correctly
    represented in LEAN code.
2. Mathematical Accuracy: The translation preserves the accuracy of the
    mathematical content.
3. Structural Fidelity: The translation aligns closely with the
    original problem, maintaining its structure and purpose.
4. Comprehensiveness: All assumptions, conditions, and goals present in
    the natural language statement are included in the LEAN translation.

Your answer should be in the following format:

Thought: [Your Answer]

Judgement: [Your Answer, one of {Appropriate, Inappropriate}]

---

The following are the example problems labeled for the reasonability of
    their translation.
```

```
# Example 1:

## Original natural language statement of the problem:

For the graph of a certain quadratic y = ax² + bx + c, the vertex of the
    parabola is (2,10), and one of the x-intercepts is (1,0).  What is the
    x-coordinate of the other x-intercept?

## Translated formal statement:
```lean
theorem quadratic_other_intercept
    (f : ℝ → ℝ)
    (a b c : ℝ)
    (h_quad : ∀ x, f x = a * x^2 + b * x + c)
    (h_vertex : f 2 = 10 ∧ ∀ x, f x ≤ f 2)   -- vertex at (2,10)
    (h_intercept1 : f 1 = 0) :                -- x-intercept at (1,0)
    f 3 = 0 := by                             -- other x-intercept at
    (3,0)
```

Thought: The Lean translation of the problem is appropriate because it
    accurately captures the intent and reasoning of the original
    problem. The problem's key elements, such as the vertex, axis of
    symmetry, and x-intercepts of the quadratic function, are correctly
    translated into Lean code. The logical flow of the proof mirrors
    the original reasoning, starting with the symmetry property of the
    quadratic function and using it to determine the second
    x-intercept. The mathematical accuracy is preserved, as the proof
    correctly applies the vertex property and symmetry to arrive at the
    solution. Furthermore, the translation aligns well with the
    original problem in natural language, maintaining fidelity to its
    structure and purpose. Overall, the translation is both faithful
    and complete, making it an appropriate representation of the
    original problem.

Judgement: Appropriate

# Example 2:

## Original natural language statement of the problem:

Draw a tangent line from the point (4,3) to the circle (x − 2)² + (y − 1)² = 1
    . What is the equation of the line that passes through the two
    tangent points?

## Translated formal statement:
```lean
theorem tangent_line_equation (x y : ℝ) :
  let P : ℝ × ℝ := (4, 3)   -- Point P
  let C : ℝ × ℝ := (2, 1)   -- Center of first circle
  let r : ℝ := 1            -- Radius of first circle
  -- Points (x,y) satisfying both circle equations
  let on_first_circle := (x − 2)^2 + (y − 1)^2 = 1
  let on_second_circle := (x − 3)^2 + (y − 2)^2 = 2
  -- If point is on both circles
  on_first_circle ∧ on_second_circle →
  -- Then it lies on the line 2x + 2y − 7 = 0
  2*x + 2*y − 7 = 0 := by
```

Thought: The Lean translation of the problem is inappropriate because
    it fundamentally changes the intent of the original problem. The
    original problem asks to derive the equation of the tangent line
```

passing through the intersection points of two circles, but the
translation assumes the equation (2x + 2y - 7 = 0) is already given
and instead asks to prove that the intersection points lie on this
line. This shifts the problem from a construction task to a
verification task, losing the original problem's focus on deriving
the result through geometric and algebraic reasoning. Additionally,
the translation omits the key reasoning step of subtracting the
circle equations to derive the line equation, which is central to
the original problem. As a result, the translation fails to
accurately represent the problem's intent and educational value,
making it an incomplete and inappropriate representation.

Judgement: Inappropriate

Example3:

## Original natural language statement of the problem:

If $a, b, c, d > 0$ and $abcd = 1$ , prove that \n\n
$\frac{1}{a+b+c+1} + \frac{1}{b+c+d+1} + \frac{1}{c+d+a+1} + \frac{1}{d+a+b+1} \leq \frac{1}{a+3} + \frac{1}{b+3} + \frac{1}{c+3} + \frac{1}{d+3}$ \n\n -/

## Translated formal statement:
```lean4
theorem lean_workbook_49553 (a b c d : ℝ) (habc : a * b * c * d = 1) :
    (1 / (a + b + c + 1) + 1 / (b + c + d + 1) + 1 / (c + d + a + 1) +
    1 / (d + a + b + 1)) ≤ (1 / (a + 3) + 1 / (b + 3) + 1 / (c + 3) + 1
    / (d + 3))  :=  by sorry
```

Thought: The Lean translation of the problem is inappropriate because
the condition $a, b, c, d > 0$ is ignored in the formal statement.

Judgement: Inappropriate

Example4:

## Original natural language statement of the problem:

If $a = b = c = 2$ so $\sum_{cyc} \frac{(a-1)^2}{a^2+2} = \frac{1}{2}$ . We'll prove that $\frac{1}{2}$ is the answer.

## Translated formal statement:
```lean4
theorem lean_workbook_plus_1478 (a b c : ℝ) (ha : a = 2) (hb : b = 2)
    (hc : c = 2) : (a - 1) ^ 2 / (a ^ 2 + 2) + (b - 1) ^ 2 / (b ^ 2 +
    2) + (c - 1) ^ 2 / (c ^ 2 + 2) = 1 / 2   :=  by sorry
```

Thought: The Lean translation of the problem is appropriate because it
accurately captures the assumptions and the goal in the natural
language statement.

Judgement: Appropriate

## Original natural language statement of the problem:

{informal_statement}

## Translated formal statement:
```lean4
{formal_statement}
```

```
```
```

# E    DETAILS FOR INFORMAL-BASED SCAFFOLDED DATA SYNTHESIS

In this section, we provide all the details for informal-based scaffolded data synthesis. We start with the prompt template for different LLM queries, including the prompt template to generate a harder variant, simpler/sub-problem, as well as the prompt template to judge the simplicity and correctness. Then, we provide more details for the synthesis pipeline.

**Prompt template for solving the original problem**    Note that for both input with natural language problem and statement written in Lean, we use the same prompt because we find that general-purpose models can understand the Lean, although they cannot generate the whole proof correctly.

```
Solve the following math problem probably written in Lean 4:
{problem}
Provide a detailed solution. Note that you don't need to prove the
    problem in Lean 4, just provide a detailed solution in natural
    language or math notation.
```

**Prompt template for generating sub/simpler problems**

```
I will give you a math problem along with its full solution.
Your task is to generate simpler problems which may enable a student to
    build up the skills to solve the given problem. Each simpler
    problem should reflect the idea of a core step in the solution.
Each generated problem must:
(1) Be completely self-contained and standalone: it should be clearly
    stated as an independent question that someone could read and work
    on without seeing the original problem and the other generated
    problems.
(2) Be purely proof-based: stated explicitly as a question of the form
    "Prove that...".
(3) Be related to the core steps in the solution of the original
    problem or reflect the core idea, and should not be just a trivial
    and straightforward derivation, plug-in calculation, or solving
    simple equations.
After generating the problems, carefully evaluate your own output and
    perform the following checks:
(1) Ensure each problem is fully self-contained: check that it does not
    rely on undefined variables, terms, or concepts from the original
    problem or other problems.
(2) Ensure the set of problems is diverse, covering different steps or
    aspects of the original reasoning.
(3) Ensure that the problems are not too simple or trivial, and that
    they require a meaningful proof. Avoid trivial and straightforward
    derivation, plug-in calculation, or solving simple equations.
Wrap each final selected problem between the tags <newproblem> and
    </newproblem> to make it easy to extract.
Do not include anything else.
Problem: {problem}
Solution: {solution}
```

**Prompt template for generating a harder variant of problems**

```
I now have a math problem and its solution at hand, and I would like
    you to modify the problem to generate a diverse set of new problems
    based on it. Below is the problem (probably written in Lean 4):
---
{problem}
---
```

```
Below is the solution:
---
{solution}
---

Now I would like you to generate at least 10 new math problems, each
    clearly different from the original. For example, you can make the
    following modifications to make the problem different:
Change the number in the original problem to generate a new problem.
Transform the algebraic formula such that it needs more simplification.
    For example, change cos(x) in the original problem into 1 - 2
    sin(x/2)^2.
Add more terms to the inequality. For example, the original problem
    needs to show that f(x) is non-negative, you transform the problem
    into f(x) + (1/x - a)^2 is non-negative, or f(x) \cdot (x - b)^2 is
    non-negative.
Lift a real variable into a complex number, a vector or even a matrix.
    For example, previously when solving a quadratic equation in real
    space, you would now change it to a complex field, which might lead
    to slightly different solutions. If the original problem considers
    planar geometry, you modify the problem into 3-dimensional geometry.
Substitute a variable into a more complex algebraic form, which
    includes but is not limited to changing a variable into a
    polynomial, exponential, logarithm, or even trigonometric form. For
    example, you change variable x in the original problem into y^2 or
    even y^n in the new problem. Another example is that you are given
    some condition like a + b + c = 1 where a, b, c are all positive,
    and you need to show f(a, b, c) is non-negative. You can change the
    condition into x y + y z + z x = x y z with x, y, z positive, which
    is equivalent to 1/x + 1/y + 1/z = 1, and modify the statement to
    show that f(1/x, 1/y, 1/z) is non-negative.
Use the conclusion in the problem as a step to solve another problem.
    For example, the original problem needs to solve the equation f(x) =
     0, where x is the variable. Now you change it to solve f( exp(x) )
    = 0, f( ln(x) ) = 0, or f( cos(x) ) = 0, or even f( x^2 ) = 0.
These are just some example transformations you can try, and you are
    not limited to these transformations. Do not overcomplicate the
    problem, and it is acceptable to make simple transformations. The
    new math problem should not be simpler than the example problems I
    provided, i.e., you should not simplify the original problem and
    make it the new problem.
Each generated problem must:
(1) Be purely proof-based: stated explicitly as a question of the form
    "Prove ...that". Do not generate problems that ask to "determine",
    "compute", "evaluate", "find", or similar.
(2) Be completely self-contained and standalone.
(3) Be mathematically valid and solvable.
(4) Be meaningfully different to ensure diversity.
Wrap each generated problem between the tags <newproblem> and
    </newproblem> to make it easy to extract.
Do not include anything else.
```

**Prompt template to judge the correctness and simplicity of the synthesized statement**

```
I will give you a math problem written in Lean 4, and I will ask you to
    determine (1) whether the problem is correct or not; and (2)
    whether the problem is too simple to prove or disprove in Lean 4.
    Here is a list of too simple problems in Lean 4: simple
    calculations, simple algebraic manipulations, solving single
    variable linear equations (by just a 1-step calculation), and
    inequalities proved by an easy sum-of-squares technique. However,
    do not include inequality proving with the square root since that
    might be more complex. Please do not label other problems, such as
```

```
      other more complex inequalities, limits, and integrals, as simple
      problems. Also, please do not label problems that deals with
      integers (more related to number theory), higher order roots,
      complex numbers, matrices, polynomials, group, finite-sum, or
      functional equations (e.g. Let f be a function such that  f(x^2) =
      xf(x)  for all  x \\in R.  Prove that f is an odd function, i.e.,
      f(-x) = -f(x)  for all  x \\in R .), since these problems might
      shed lights on other hard problems.
Please carefully analyze the problem and provide an explanation of your
      reasoning.
For judging the correctness, if the problem is correct, respond with
      "yes"; if the problem is incorrect, respond with "no"; if you are
      not sure, respond with "unsure".
For judging the simplicity, if the problem is too simple to prove or
      disprove in Lean 4, respond with "yes"; if the problem is not too
      simple, respond with "no"; if you are not sure, respond with
      "unsure". Please refer to the list of too-simple problems I
      provided.
Wrap your judgment between <judge> and </judge> to make it easy to
      extract. You should first answer yes or no to the correctness and
      then answer yes or no to the simplicity, separated by a comma. For
      example, if you think the problem is correct and not too simple,
      you should respond with <judge>yes, no</judge>. If you are not sure
      about the correctness, you can respond with <judge>unsure,
      no</judge> because at least this problem is not simple. If you
      think the problem is incorrect but it is not too easy to disprove,
      you can respond with <judge>no, no</judge>. If you think the
      problem is correct but too simple to prove or disprove, you can
      respond with <judge>yes, yes</judge>. Again, when judging the
      simplicity, please do not label inequality proving with the square
      root as simple, since that might be more complex than expected.
      Please do not include problems such as more complex inequalities,
      or problems that include limits, and integrals. Also, please do not
      label problems that deals with integers (more related to number
      theory), higher order roots, complex numbers, matrices,
      polynomials, group, finite-sum, or functional equations, as simple
      problems, since these notions are more likely to related to hard
      problems and proving problems related to these concepts in Lean 4
      might not be easy even if the problem is straight-forward in
      natural language.
Do not include anything else.
Problem: {formal_statement}
```

**More details for informal-based scaffolded data synthesis**   In the following part, we discuss some of the design choices and details for the scaffolded data synthesis pipeline.

1. **(Informal statements generation.)** For the informal statements generation, one design choice is whether to generate multiple questions during the same inference, or to generate multiple times while only generating 1 (or very few) questions during one inference. From our experiment, we observe that for the Qwen3-32B model, generating multiple questions during the same inference is better, since the generated questions are likely to be different. Otherwise, there might be very similar questions among different generations. Besides, we also find that repeatedly generating a single problem multiple times doesn't significantly increase the number of different problems. Thus, for efficiency considerations, we only query the LLM (Qwen3-32B) once for generating multiple hard variants/simple problems of a given math problem.

2. **(Formalization and quality checking.)** For each generated informal statement, we call the trained formalizer to formalize the problem twice. Then, we then query Qwen3-8B for 3 times for each formalization, and decide if the formalization is aligned with the informal statement using majority voting (among three queries). We decide to formalize each informal statement twice in order to balance the efficiency and the number of generated problems. We only keep at most one formalization for each generated informal statement.

3. **(Negation and difficulty filtering.)** For each formalization, we query Qwen3-32B for 4 times, where each inference judges the correctness and the simplicity simultaneously. The final correctness is judged by strict majority voting among the 4 judges, while the final simplicity is determined if all 4 judges think the problem is easy. We use such strict criteria to minimize the probability of discarding hard and valuable problems. The efficiency for the judging is high, even if we call Qwen3-32B 4 times for each formalization, since a lot of the time Qwen3-32B enters the "fast thinking" mode (no long chain-of-thought for reasoning).

4. **(Final deduplication.)** At the end of the pipeline, we filter out duplicated statements under exact match.

## F  RL TRAINING DETAILS

We further explain our RL training in detail.

### F.1  RL IMPLEMENTATION

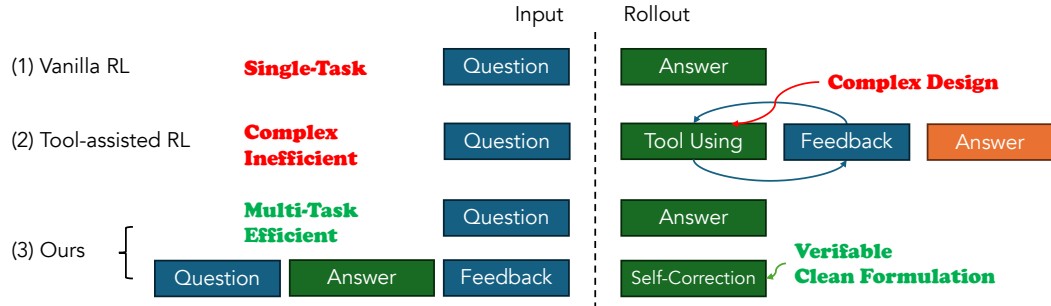

Figure 9: Illustrative figure for our multi-task RL. This pipeline improves models' performance on both the whole proof and the self-correction at the same time without additional design on the framework or algorithm.

We begin by collecting 50K challenging statements and 50K self-correction samples. Each self-correction sample contains a statement, an output generated by the SFT model, and an associated error message. During RL training, we consumed approximately 46K and 64K unique inputs for the 8B and 32B models on 1 epoch, respectively. We use the VeRL framework (Sheng et al., 2024) with several key setups: We adopt a batch size of 128 and n of 8 for parallel rollouts and reward function calls (via the Lean compiler), while using a mini-batch size of 32, accepting a certain degree of off-policy training in exchange for a higher frequency of policy optimization. For the dynamic sampling, we set the over-sample batch size equal to three times of train batch size and filter out inputs with a pass rate equal to 0 or higher than 0.75. We use a maximum prompt length of 16K for those long inputs in the self-correction task. We set the maximum response length to 24K to support a reasonably sized reasoning trajectory while staying within the Qwen3 model's native 40K context window, ensuring generation quality. We enable the overlong penalty with a 4K overlong buffer and set the overlong penalty factor to 1. We use token-averaged policy loss and do not use KL divergence or entropy terms in our training objective.

### F.2  FURTHER DISCUSSION ON RL

For the design of training with self-correction, we explored both tool-use and multi-turn reinforcement learning. However, the former not only requires a dedicated tool-calling design but also demands strong model capability to follow the tool-calling protocol, especially in complex and lengthy formal language scenarios like Lean, which is particularly challenging for our relatively small model. Meanwhile, multi-turn approaches introduce various engineering challenges, especially on the rollout engine side, such as asynchronous generation. Moreover, on the algorithm side, the effectiveness and efficiency of multi-turn RL still require further validation. We have explored some preliminary approaches, but they remain immature. Therefore, we adopt a more straightforward implementation

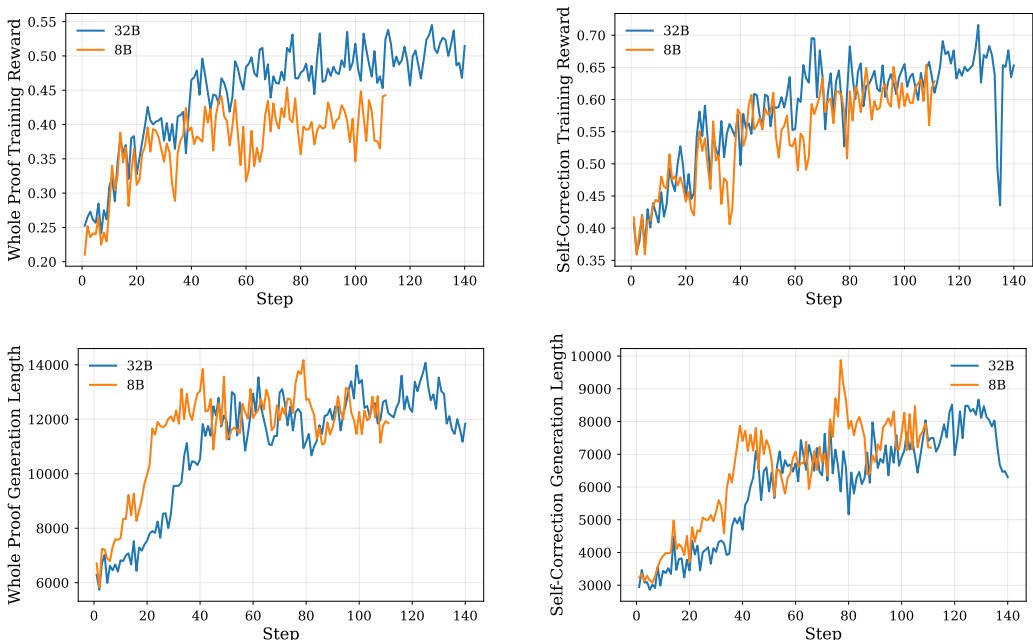

Figure 10: Training curves comparing 32B and 8B models: training rewards (top row) and generation lengths (bottom row) for whole proof completion and self-correction tasks. Note that the rewards are averaged over all generated rollouts, including those then filtered by the dynamic sampling strategy, thus reflecting the policy improvement.

that is natively integrated into the current RL framework, as illustrated in Figure 9. In this setup, we have two different types of inputs for RL, and detailed training curves are shown in Figure 10. Furthermore, for the algorithmic design, like the advantage estimator, we explored different popular GRPO variants but did not observe significant differences.

## G    PROOFNET AND RETRIEVAL-AUGMENTED GENERATION.

ProofNet (Azerbayev et al., 2023) focuses on undergraduate-level pure mathematics, with problems drawn primarily from standard undergraduate textbooks covering subjects such as real and complex analysis, linear algebra, abstract algebra, and topology. Notably, ProofNet is constructed based on mathlib and primarily tests a model's ability to memorize and utilize the tactics from mathlib. The experimental results, as shown in Table 7, show that our model does not perform as well as DeepSeek-Prover-V2 on ProofNet.

To explore further, we implement a Mathlib RAG (Retrieval-Augmented Generation), where we use gpt-oss-120b model to generate informal quesies for LeanSearch (Gao et al., 2024b). We test the methods on both MiniF2F and ProofNet. Our results show that Mathlib RAG does not help, and may even hurt model's performance on MiniF2F, but it does improve results on ProofNet by a large margin, achieving roughly a 6-point accuracy gain and reaching the performance of DeepSeek-Prover-V2-671B.

This demonstrates that ProofNet relies more on leveraging complex capabilities from mathlib than on cognitive reasoning skills. ProofNet involves a distribution shift and requires advanced mathematical knowledge typically found in Mathlib; while our model does not natively integrate Mathlib well, augmenting it with Mathlib retrieval yields strong gains.

Table 7: Comparison of different methods on ProofNet and MiniF2F under pass@32.

| Method | ProofNet | MiniF2F |
|---|---|---|
| Goedel-Prover-V2-32B | 22.6% | 88.1% |
| Goedel-Prover-V2-32B-RAG | 28.5% | 86.5% |
| DeepSeek-Prover-V2-7B | 23.0% | 75.6% |
| DeepSeek-Prover-V2-671B | 30.5% | 82.4% |

