# OpenReview forum: "Goedel-Prover-V2: Scaling Formal Theorem Proving with Scaffolded Data Synthesis and Self-Correction"
_ICLR.cc/2026/Conference — ICLR 2026 Poster_

### Official Review · Reviewer_nJcq · 2025-10-15

**Soundness:** 3
**Presentation:** 3
**Contribution:** 3
**Rating:** 6
**Confidence:** 4

**Summary:**

The authors present Goedel-Prover-V2, a family of two models finetuned for automated theorem proving with Lean. The models were trained through a series of expert iteration and RL, starting with an initial seed generated by DeepSeek-Prover-V2. Additionally, the models were specifically finetuned to be able to incorporate Lean feedback and underwent a weight averaging scheme to increase diversity in the generations of the models. The results indicate that the models outperform prior open-weight alternatives on popular benchmarks like PutnamBench and Mini-F2F.

**Strengths:**

- The paper present two new models that outperform prior state-of-the-art in an important field of mathematical applications for LLMs (namely, automated proof generation).
- The training scheme (weight averaging, expert iteration with RL for Lean feedback, ...) is interesting and novel.
- Goedel-Formalizer-V2 also outperforms prior formalizers.
- Comparison with prior work is solid and appropriately uses the two most popular benchmarks in this field for the comparison.

**Weaknesses:**

I am currently giving a 6, since I believe that while the paper presents a strong contribution, there are significant areas of improvement, especially regarding the paper. Most of these weaknesses can be improved during the rebuttal phase, and I therefore see myself increasing my score if they are appropriately addressed. However, for some weaknesses it could also be that their answer will indicate further problems with the paper (especially regarding contamination and the auto-formalizer). If this is the case, I will decrease my score.

I have split the weaknesses in "weaknesses" and "remarks", where "remarks" are smaller points that only affected my rating in a minor way.

**Weaknesses**
- It is unclear why MathOlympiadBench is included. The results on this benchmark are only given in Figure 1 for three models and are never even discussed in the experimental section. I do agree that the three examples in the Appendix indicate that MathOlympiadBench is better formalized, but it is unclear what proportion of MiniF2F this constitutes and how difficult it was to find such examples. Furthermore, MathOlympiadBench has the distinct disadvantage that its theorems might have been included in training data of other models, since decontamination procedures for these models might not have included the non-overlapping problems with MiniF2F. Overall, I think the inclusion of MathOlympiadBench in its current state decreases the contribution of the paper, as it takes away the focus from Goedel-Prover-V2 and is missing a lot of details regarding experimentation, argumentation why it is necessary to have such a third dataset, and missing explanations regarding its construction (Appendix A is very unclear, as I have no idea how the issues occured, what kind of human evaluation was done, etc.). The authors should either remove the benchmark or make it a complete part of their paper by adding these details.
- The comparison between feedback and no feedback version of the models is misleading. pass@n with 2 rounds of feedback allow the model to have $3n$ attempts at the problem, not $n$. As such, comparisons should be made between pass@n of the feedback setting with pass@3n of the non-feedback setting. I do note that this is a back-of-the-envelope calculation, as feedback rounds might spend less compute tokens (but it does have more input tokens), but at least the FLOPs between these settings should be matched for proper comparisons. If I look at the numbers, the feedback settings helps much less when one does this more proper comparison (but one cannot entirely tell from the numbers included in the paper alone).
- It is not mentioned how the results in Table 1 were obtained. If, as I suspect, correctness was verified by the same LLM judge, how can we ensure that the Goedel-Formalizer did not simply learn to fool this LLM judge better? It was specifically finetuned to do better according to this LLM judge, automatically leading to more true positives and false positives. If these results were indeed obtained using the same LLM judge, it would be required to perform another (mabye smaller-scale) evaluation method to compare the formalizers.
- It is never mentioned where the training data problems were sourced from. Furthermore, it is never mentioned how decontamination with respect to the test benchmark was performed.
- Currently, no code is included. It would be appreciated if code could be included to verify some of the statements made etc.

**Remarks**
- I disagree that including CoT in Figure 7 has any influence on performance. The two lines are clearly within 1 standard deviation of each other and the difference is overall very minimal. Furthermore, including the CoT also leads to more compute that needs to be spent per task, indicating that it is likely the opposite case: It is probably better to not include the CoT because it is basically the same performance for less compute.
- In Figure 8, it is unclear why we are looking at checkpoints rather than the final model. Also, the differences between the different $\alpha$'s are tiny, and much less clear-cut than what is discussed in the text.
- I find Section 2 confusing in places. Specifically, the data generation pipeline is very unclear. It should be more clearly mentioned how:
  - How the data generation pipeline with scaffolding is used in each stage (is it used all the time, only in some stages, when is it used, how is the data generated to check whether it can be solved, ...).
  - Where the problems were originally sourced from.
  - Specifically, 2.4 only ever mentions what happens when you have the formalized statements ready, but it is not clear how these formalized statements are obtained in each stage due to the missing details above.
- The motivation in the first paragraph of the introduction feels a bit off (especially L45-47): Kimina-Prover has the same number of parameters as Goedel-Prover-V2 so the argument does not really hold there. The motivation is much simpler than what presented there ("we want to create even better models"). It would be better to incorporate this argument instead.
- Kimina-Prover should be included in Fig 1 center and right as well.
- In L98: "ultimately bridge the long-standing divide between intuitive human reasoning and formal proof verification." -> This feels a bit too grand. To do such a thing, one would at the very least need an informalizer as well, or somehow incorporate informal proofs in the formal verification process. I would remove this sentence.
- L134-136 either needs a citation or an explanation how this evaluation was performed.
- It is never mentioned which LLM is used to verify formalizations, not even in Appendix C.
- The caption of Table 1 should be above the table.
- There is an asterix in the caption of Table 2, but it is never explained why it is there.
- It is never mentioned which $\alpha$ was chosen for the final model.
- The formatting consistency needs to be improved. In one section, the paper uses bullet points without bolded headers, in another it uses bullet points with bolded headers, or just no bullet points and no bolded headers, or just bolded headers. Some of these bolded headers are capitalized, others are not, ... Plots do not have a consistent format. The discrepancy is especially prevalent between Figure 6 and 7 (different background, label size, title vs no title, ...).
- The writing in the appendix D needs a pass, it contains several typos and grammatical mistakes.

**Questions:**

See weaknesses

---

> ### Author Response · Authors · 2025-11-22
> **Reply to Reviewer nJcq (1/4)**
>
> Thank you so much for your detailed review! We value your suggestions address your concerns below.
>
> ---
>
> ## The Inclusion of MathOlympiadBench
>
> > It is unclear why MathOlympiadBench is included. The results on this benchmark are only given in Figure 1 for three models and are never even discussed in the experimental section.
>
> We clarify in Section 3.1 that we evaluate our models on this dataset, and we also provide a pointer to it in the first paragraph of Section 3.3. In our original experiments, we reported results for only the strongest models on MathOlympiadBench, following a similar approach to DeepSeek-Prover-V2's evaluation on DeepSeek-ProverBench, since the difficulty of this dataset makes it an informative but demanding metric.
>
> To provide a more comprehensive comparison, we have now included additional models in our evaluation. The table below reports the number of problems solved (pass@32) out of 360 MathOlympiadBench questions for each model, the setting is the same as our main experiments.
>
> | DeepSeek-Prover-V2-7B | DeepSeek-Prover-V2-671B | Kimina-Prover-8B-Distill | Kimina-Prover-72B |
> | :---- | :---- | :---- | :---- |
> | 28 | 50 | 29 | 37 |
> | **Goedel-Prover-V2-8B** | **Goedel-Prover-V2-8B w/ revision** | **Goedel-Prover-V2-32B** | **Goedel-Prover-V2-32B w/ revision** |
> | 37 | 40 | 60 | 73 |
>
> As shown, our models maintain a strong overall lead on MathOlympiadBench. Our 32B model achieves the best performance among all models evaluated. The lower performance of our 8B model compared to DeepSeek-Prover-V2-67B is expected due to the significant difference in model size. These results highlight the robustness and competitiveness of our approach, even on the most challenging mathematical tasks.
>
> > I do agree that the three examples in the Appendix indicate that MathOlympiadBench is better formalized, but it is unclear what proportion of MiniF2F this constitutes and how difficult it was to find such examples.
>
> Thank you for pointing this out\! We have manually examined all overlapping problems between MiniF2F test and MathOlympiadBench, and have revised Appendix A to explicitly clarify this.
>
> > Furthermore, MathOlympiadBench has the distinct disadvantage that its theorems might have been included in training data of other models, since decontamination procedures for these models might not have included the non-overlapping problems with MiniF2F.
>
> While problems in MathOlympiadBench are sourced from IMO and IMO short lists, we should note that MiniF2F contains problems partially from IMO, AIME, and AMC, and DeepSeek-ProverBench includes problems in part from AIME.
>
> MathOlympiadBench faces the same risk of data contamination as these datasets. Notably, recent works on informal mathematics also widely use historical competition problems like AIME for evaluation. In our own training process, we have strictly filtered the training data to ensure there is no overlap with the test set and to avoid any data leakage. If someone were to deliberately attempt to induce data contamination, our dataset would face similar risks as other related datasets, which is out of the scope of this work.
>
> > Overall, I think the inclusion of MathOlympiadBench in its current state decreases the contribution of the paper, as it takes away the focus from Goedel-Prover-V2 and is missing a lot of details regarding experimentation, argumentation why it is necessary to have such a third dataset, and missing explanations regarding its construction (Appendix A is very unclear, as I have no idea how the issues occurred, what kind of human evaluation was done, etc.). The authors should either remove the benchmark or make it a complete part of their paper by adding these details.
>
> MathOlympiadBench is not the primary contribution of our work, but it is nonetheless important for the community, and it is also valuable in demonstrating the capabilities of Goedel-Prover-V2. Many recent works, such as AlphaProof, claim IMO-level proficiency, yet there are no publicly available IMO-level datasets. Our dataset addresses this gap. Thank you for raising these points, and we have now added the motivation for including this dataset to the paper.
>
> It is also worth noting that there is an overall scarcity of high-quality datasets in this field. As a result, many recent works, such as Kimina-Prover and DeepSeek-Prover-V2, have included new test datasets together with their models. We believe that MathOlympiadBench is aligned with this broader community effort to increase the rigor of benchmarks available for formal theorem proving.

---

> ### Author Response · Authors · 2025-11-22
> **Reply to Reviewer nJcq (2/4)**
>
> ## Comparisons Between Feedback and No-Feedback Settings
>
> > The comparison between feedback and no feedback version of the models is misleading. pass@n with 2 rounds of feedback allow the model to have attempts at the problem, not . As such, comparisons should be made between pass@n of the feedback setting with pass@3n of the non-feedback setting. I do note that this is a back-of-the-envelope calculation, as feedback rounds might spend less compute tokens (but it does have more input tokens), but at least the FLOPs between these settings should be matched for proper comparisons. If I look at the numbers, the feedback settings helps much less when one does this more proper comparison (but one cannot entirely tell from the numbers included in the paper alone).
>
> Thank you for raising this point! I understand your concern. In our main experiments, the value of *n* refers to the number of system runs, not the total number of attempts per problem. When considering computational cost in terms of FLOPs, a commonly used estimate in the community is $\text{model size} \times \text{output length}$ \[1\]. As described in Section 3.2, we limit the maximum context length in the main experiments, so the total computational cost does not simply scale by 3× with two rounds of revision compared to the vanilla setting.
>
> To provide a more precise comparison, we have computed the average **total** output token lengths for the vanilla (no feedback), 1-round revision, and 2-round revision settings:
>
> | Model | Vanilla | Revision 1 | Revision 2 |
> | ----- | ----- | ----- | ----- |
> | Goedel-Prover-V2-8B | 7,641.9 | 8,551.0 | 8,878.0 |
> | Goedel-Prover-V2-32B | 6,847.7 | 7,587.2 | 7,978.2 |
>
> As shown, two rounds of revision increase the average output length by less than 20% compared to the vanilla setting, rather than by a factor of three. We have added these findings to our manuscript for clarity. Additionally, we have visualized the relationship between performance and output length and included this analysis in the paper. The results demonstrate that the revision approach utilizes the compute more efficiently.
>
> ## Autoformalization Validity and Reward Hacking
>
> > It is not mentioned how the results in Table 1 were obtained. If, as I suspect, correctness was verified by the same LLM judge, how can we ensure that the Goedel-Formalizer did not simply learn to fool this LLM judge better? It was specifically finetuned to do better according to this LLM judge, automatically leading to more true positives and false positives. If these results were indeed obtained using the same LLM judge, it would be required to perform another (mabye smaller-scale) evaluation method to compare the formalizers.
>
> > L134-136 either needs a citation or an explanation how this evaluation was performed.
>
> > It is never mentioned which LLM is used to verify formalizations, not even in Appendix C.
>
> We thank the reviewer for raising these critical points.
>
> The results in Table 1 were obtained using the automated judge described in Section 2.2.1 and Appendix D, which the same strict pipeline used during expert iteration. We have added this description in the revised paper.
>
> Specifically, our semantic correctness check employs a **strict consensus mechanism**: we query the judge (Qwen3-32B) four times independently for each statement. A formalization is marked as correct *only* if all four queries unanimously agree that it is faithful.
>
> Regarding ‘Fooling the Judge’, we have explicitly anticipated the risk of the model learning to "fool" the formalizer. We implemented two safeguards to mitigate this:
>
> - **Limited Iterations:** We restricted the training to only two rounds of expert iteration. This prevents the mode collapse and reward hacking often observed in prolonged training loops.
> - **Human Validation:** To empirically verify the judge's robustness, we conducted a human evaluation on 30 sampled problems. We compared the "unanimous consensus" (4/4) LLM judgments against expert human annotations. This audit revealed a **92% agreement rate** between the automated judge and human experts. We believe this high alignment demonstrates that our results reflect genuine performance improvements rather than reward exploitation.
>
> \[1\] Scaling LLM Test-Time Compute Optimally can be More Effective than Scaling Model Parameters, ICLR 2025

---

> ### Author Response · Authors · 2025-11-22
> **Reply to Reviewer nJcq (3/4)**
>
> ## Sources of Training Problems and Data Pipeline Clarity
>
> > It is never mentioned where the training data problems were sourced from.
>
> > I find Section 2 confusing in places. Specifically, the data generation pipeline is very unclear. It should be more clearly mentioned how: How the data generation pipeline with scaffolding is used in each stage (is it used all the time, only in some stages, when is it used, how is the data generated to check whether it can be solved, ...). Where the problems were originally sourced from. Specifically, 2.4 only ever mentions what happens when you have the formalized statements ready, but it is not clear how these formalized statements are obtained in each stage due to the missing details above.
>
> The training data primarily comes from Goedel-Pset, STP, the scaffolded data sources mentioned in our paper, as well as from our own formalization of the Nvidia/OpenMathReasoning dataset using our newly trained formalizer.
>
> We have now clarified these sources and rewritten the pipeline description in Section 2.4 and revised Figure 3 to make the data pipeline more transparent.
>
> ## Decontamination
>
> > Furthermore, it is never mentioned how decontamination with respect to the test benchmark was performed.
>
> Regarding decontamination, following the practice of previous work, we compared the formal statements in the training and test sets to prevent data leakage. We have added this description to our paper.
>
> ## The Availability of Code
>
> > Currently, no code is included. It would be appreciated if code could be included to verify some of the statements made etc.
>
> Thank you for your suggestion! We have now uploaded supplementary materials, including our inference code and model averaging code. The SFT and RL training code is based on open-source frameworks, which is also stated in the paper.
>
> ## Model Averaging Hyperparameter
>
> > It is never mentioned which $\alpha$ was chosen for the final model.
>
> We used an $\alpha = 0.8$ for all model averaging. We have added it to Section 2.3 of the paper.
>
> ## Experiments on Model Averaging Hyperparameter and RL
>
> > In Figure 8, it is unclear why we are looking at checkpoints rather than the final model. Also, the differences between the different 's are tiny, and much less clear-cut than what is discussed in the text.
>
> We would like to clarify that our experiments include not only multiple RL checkpoints but also the final model. For each checkpoint and the final model, we applied model averaging with different values of $\alpha$ to systematically examine the effects of both the RL training step and the averaging ratio. For every setting, we evaluated at least pass@64 and then reported the averaged pass@32 and pass@1 scores. As a result, the observed differences are robust and not due to random fluctuation.
>
> We observe two clear trends: (1) as the RL step increases, pass@1 improves while pass@32 fluctuates; (2) as the base model ratio increases, pass@1 consistently decreases, while pass@32 first increases and then decreases. These trends indicate that RL training primarily benefits top-1 performance, while model averaging enhances diversity and therefore pass@32. This systematic analysis demonstrates the complementary effects of RL progression and model averaging, and highlights the value of model averaging for improving solution diversity.

---

> ### Author Response · Authors · 2025-11-22
> **Reply to Reviewer nJcq (4/4)**
>
> ## Effectiveness of CoT in history messages
>
> > I disagree that including CoT in Figure 7 has any influence on performance. The two lines are clearly within 1 standard deviation of each other and the difference is overall very minimal. Furthermore, including the CoT also leads to more compute that needs to be spent per task, indicating that it is likely the opposite case: It is probably better to not include the CoT because it is basically the same performance for less compute.
>
> Thank you for raising this question. We agree with your assessment and have updated our analysis accordingly to reflect that including CoT does not provide a significant performance benefit, while incurring additional computational cost.
>
> ## Writing Issues
>
> > The motivation in the first paragraph of the introduction feels a bit off (especially L45-47): Kimina-Prover has the same number of parameters as Goedel-Prover-V2 so the argument does not really hold there. The motivation is much simpler than what presented there ("we want to create even better models"). It would be better to incorporate this argument instead.
>
> > In L98: "ultimately bridge the long-standing divide between intuitive human reasoning and formal proof verification." \-> This feels a bit too grand. To do such a thing, one would at the very least need an informalizer as well, or somehow incorporate informal proofs in the formal verification process. I would remove this sentence.
>
> > The caption of Table 1 should be above the table.
>
> > There is an asterix in the caption of Table 2, but it is never explained why it is there.
>
> > The formatting consistency needs to be improved. In one section, the paper uses bullet points without bolded headers, in another it uses bullet points with bolded headers, or just no bullet points and no bolded headers, or just bolded headers. Some of these bolded headers are capitalized, others are not, ... Plots do not have a consistent format. The discrepancy is especially prevalent between Figure 6 and 7 (different background, label size, title vs no title, ...).
>
> > The writing in the appendix D needs a pass, it contains several typos and grammatical mistakes.
>
> Thank you for your careful reading and valuable suggestions. We have revised our writing accordingly based on your feedback.
>
> ---
>
> Thank you again for your time and effort! We hope our response resolves your concerns. We welcome any further comments.

---

> > ### Comment · Reviewer_nJcq · 2025-11-24
> > **Reply to Rebuttal**
> >
> > # Comparisons Between Feedback and No-Feedback Settings
> > I thank the authors for their inclusion of these results. However, I find Fig 6b a lot less favorable for the revision method than discussed in the paper, and than Fig 6a indicates. Its likely related to the figure starting with the equivalent of pass@1, but there the revisions barely seem to help (and are even under the line). Could the authors take a guess why this happens? Also, could they compute a deviation similar to Fig 7 to see whether the difference between revisions and no revisions is actually significant? By eyeballing it a bit, it basically seems ineffective (maybe a bit?).
> >
> > Generally, given that so much of the training focuses on correcting revisions, it is somewhat disappointing that this only improves performance marginally. Yet, in the paper it is made seem as some big achievement, while I think the takeaway here might, critically, be the opposite: it is really not worth spending half your compute budget for. I would find it somewhat surprising if that compute was instead spent on just attempting to improve the model, it would not have obtained a similar (small) performance boost. At least, it deserves a somewhat more critical analysis than what is currently shown in the paper.
> >
> > Also, the current comparison with FLOPs does not take into account the long time it usually takes for Lean to compile the proofs. I know from experience that this can take up a significant portion of time (although this observation was made with respect to a previous generation of models that didn't think that long). Could the authors check whether Lean compilation takes up a significant portion of the inference time? If so, I guess we should correct the plots even further?
> >
> > # Autoformalization Validity and Reward Hacking
> > I accept the points made here.
> >
> > # Sources of Training Problems and Data Pipeline Clarity
> > This makes things a lot more readable, thank you. I did notice a couple of typos in the revision, so it might be worth using a tool like Grammarly to fix mistakes for the camera-ready (just a side remark though).
> >
> > # Decontamination
> > It seems like you have not added any details about this in the paper (Ctrl+F on contamination does not give anything). Given the data sources, this point has actually become much, much more important: both datasets used contain problems from AOPS, which is a forum that contains basically all IMO, AIME, AMC, and Putnam problems (basically all sources for the benchmarks on which evaluation takes place). A simple decontamination check between formalizations is for sure not sufficient to fully decontaminate this data...
> >
> > # The Availability of Code
> > Thank you.
> >
> > # Model Averaging Hyperparameter
> > Thank you.
> >
> > # Experiments on Model Averaging Hyperparameter and RL
> > Ok, I think L412-415 confused me then: it never explicitly mentions that model averaging is performed between the RL checkpoint and the final model. In retrospect, this makes sense (not sure how I originally interpreted this), but might make it clearer to explicitly add this.
> >
> > # Effectiveness of CoT in history messages
> > Thank you.
> >
> > # Writing Issues
> > Thanks, I do still somewhat disagree with the formulation in L95-97: how does Goedel-Prover-v2 make significant steps towards enabling this "alignment"? As far as I can see, it still produces proofs in formal language, requires input in formal languages, and does not significantly increase this alignment with respect to prior work.

---

> ### Author Response · Authors · 2025-11-26
> **Follow-up Reply - 1/2**
>
> Thank you for acknowledging our improvements and clarifications regarding data source transparency, data pipeline description, code availability, and model hyperparameter choices in your previous round of feedback. We address your new questions below.
>
> ---
>
> ## Comparisons Between Feedback and No-Feedback Settings
>
> > I thank the authors for their inclusion of these results. However, I find Fig 6b a lot less favorable for the revision method than discussed in the paper, and than Fig 6a indicates. Its likely related to the figure starting with the equivalent of pass@1, but there the revisions barely seem to help (and are even under the line). Could the authors take a guess why this happens? Also, could they compute a deviation similar to Fig 7 to see whether the difference between revisions and no revisions is actually significant? By eyeballing it a bit, it basically seems ineffective (maybe a bit?).
>
> > Generally, given that so much of the training focuses on correcting revisions, it is somewhat disappointing that this only improves performance marginally. Yet, in the paper it is made seem as some big achievement, while I think the takeaway here might, critically, be the opposite: it is really not worth spending half your compute budget for. I would find it somewhat surprising if that compute was instead spent on just attempting to improve the model, it would not have obtained a similar (small) performance boost. At least, it deserves a somewhat more critical analysis than what is currently shown in the paper.
>
> Yes! As you pointed out, our previous figure started from pass@1, which made the effect of revision less clear at low n. Following your suggestion, we have redrawn the curve starting from pass@8 and included results for up to 128k context length and 5 rounds of revision. In the new figure, we also mark key values with dashed lines.
>
> With these changes, the role of revision is more visible: for the *same* total output token budget, the revision curves consistently lie above the no‑revision curves.
>
> Concretely, for the *2‑round revision* setting, while the average output length increases by \< 20% compared to the vanilla (no‑revision) setting, the pass@32 performance matches what the vanilla setting achieves by roughly doubling the number of passes/tokens.
>
> In our *extended experiments* with 128k context and 5 rounds of revision, the effect is even clearer. The model reaches **an average pass@32 of 92.8% and a maximum of 93.0%**. By contrast, the vanilla no‑correction setting only reaches **a maximum of 92.2% at pass@8192**. The revision setting reaches better accuracy with about 1/218 of the compute of the vanilla pass@8192 regime.
>
> This gap is much larger than the two settings from the old version of Figure 7, where the mean difference between the two curves was roughly 0.1. The new analysis and visualization make it clear that, when comparing on a fair compute budget (in tokens) and looking beyond the very low‑pass regime, revision delivers substantial gains.
>
> We agree with you that the previous figure understated this effect, and have updated both the plots and the discussion in the paper to reflect a more accurate and critical comparison.
>
> > Also, the current comparison with FLOPs does not take into account the long time it usually takes for Lean to compile the proofs. I know from experience that this can take up a significant portion of time (although this observation was made with respect to a previous generation of models that didn't think that long). Could the authors check whether Lean compilation takes up a significant portion of the inference time? If so, I guess we should correct the plots even further?
>
> In our environment, **Lean compilation accounts for only a small fraction of wall‑clock time.** For MiniF2F with Goedel‑Prover‑V2‑32B at pass@32, using 4×H100 GPUs, the inference time is about **6 hours** for the first round generation, while Lean compilation of all generated proofs takes about only **20 minutes**.

---

> ### Author Response · Authors · 2025-11-26
> **Follow-up Reply - 2/2**
>
> ## Decontamination and AOPS‑Related Concerns
>
> > It seems like you have not added any details about this in the paper (Ctrl+F on contamination does not give anything). Given the data sources, this point has actually become much, much more important: both datasets used contain problems from AOPS, which is a forum that contains basically all IMO, AIME, AMC, and Putnam problems (basically all sources for the benchmarks on which evaluation takes place). A simple decontamination check between formalizations is for sure not sufficient to fully decontaminate this data...
>
> You are absolutely right that searching for “decontamination” will not find anything in the PDF. In our previous revision we only added the sentence: *“We compare the formal statements in the training set and the test set to avoid data leakage.”* We have now made this discussion a single paragraph with more details.
>
> Regarding your concern about AoPS as a data source, actually, the informal sources we rely on (such as OMR) have already undergone rigorous decontamination against standard mathematical benchmarks in the informal domain. Since formal benchmarks like MiniF2F are largely derived from the same families of informal problems (e.g., MATH, AoPS‑sourced competition problems), this provides a strong first layer of protection against trivial leakage via reusing the same informal questions.
>
> Moreover, following standard practice in recent work \[1\], we additionally perform exact‑string matching between formal statements in our training data and the test benchmarks, and remove any overlaps. This ensures there is no exact formal duplication between training and evaluation.
>
> On top of that, our formalized datasets contain only statements (no labels/proofs), further reducing risk.
>
> We appreciate your insistence on this point\! Now the paper is clearer and more rigorous because of it.
>
> ## Writing and Positioning
>
> > I do still somewhat disagree with the formulation in L95-97: how does Goedel-Prover-v2 make significant steps towards enabling this "alignment"? As far as I can see, it still produces proofs in formal language, requires input in formal languages, and does not significantly increase this alignment with respect to prior work.
>
> Thank you for your suggestion. We have revised the relevant sentences based on an article \[2\] from a mathematician (AMS Fellow) discussing formal and informal math.
>
> ---
>
> Thank you again for your time and effort. We believe the paper is substantially stronger as a result of your comments. If our response resolves your concerns, we would appreciate it if you could reflect it in your evaluation. We welcome any further comments.
>
> \[1\] STP: Self-play LLM Theorem Provers with Iterative Conjecturing and Proving, ICML 2025
>
> \[2\] The Shape of Math To Come

---

> > ### Comment · Reviewer_nJcq · 2025-11-26
> >
> > I think there might be a bit of a misunderstanding here. By now removing the pass@1 information, it seems to me that you are essentially actively hiding information that is useful for the interpretation of the numbers, just because it does not really match your conclusion?
> >
> > With the extended reasoning, it is possible that this conflates things a bit? If the model without revisions would have been given the extended reasoning, would it also have sometimes gone above the token budget, thereby having a lower performance? If this is not the case, I agree that this at least shows a somewhat better improvement that is more worth it, but this needs to be verified.
> >
> > My remark is more on the critical side (and somewhat unadressed in the comment by the authors): These performance gains are still quite small (especially compared with the performance gains over prior models), but you spent half your training on making sure that the model could do the revisions well. Therefore, this somewhat suggest to me that it was not really worth it. More specifically, given the knowledge you have now, would you really spend half your compute budget again on the revisions, or rather on just improving the model itself?
> >
> > Thanks for clarifying the Lean comment, that is addressed now.
> >
> > Regarding the data contamination, could the authors do a simple fuzzy string match checking between the training and test data (informal statements)? This should be relatively easy to setup and (in my opinion) should have been done all the way before training given its simplicity.
> >
> > PutnamBench has no equivalent in informal mathematics. I know no popular informal math benchmark that contains samples from the same sources as MiniF2F (IMO < 2023, AIME < 2023, AMC). Additionally, NuminaMath (on which Goedel-Pset is based) does not do any decontamination. Therefore, I think this reasoning is invalid.
> >
> > While the authors do not give solutions, they do benefit in three ways from contamination: (1) distilling from Deepseek essentially teaches the model all samples Deepseek knows, (2) any model update that reduces performance on a sample gets fixed once that sample gets trained on, (3) the data is repeated multiple times.

---

> > > ### Author Response · Authors · 2025-11-27
> > > **Follow-ip Reply - 2 - 1/2**
> > >
> > > Thank you for your continued thoughtful feedback and for helping us clarify these important points. We address your latest concerns in detail below.
> > >
> > > ---
> > >
> > > ## On pass@1 information and revision evaluation
> > >
> > > > I think there might be a bit of a misunderstanding here. By now removing the pass@1 information, it seems to me that you are essentially actively hiding information that is useful for the interpretation of the numbers, just because it does not really match your conclusion?
> > >
> > > We apologize for any misunderstanding. We did not intend to hide any results. In fact, our model with revision achieves strictly better pass@1 than without revision; however, as you pointed out previously, the scale of improvement at pass@1 is very small and essentially invisible in the plot. This is why we removed it from the main figure, not to obscure results, but to avoid misleading visual emphasis. This is also consistent with evaluation practice in the field, where pass@32 is the standard starting point.
> > >
> > > As for whether to include it or not, we are open to following your recommendation.
> > >
> > > ## On extended reasoning and token budget
> > >
> > > > With the extended reasoning, it is possible that this conflates things a bit? If the model without revisions would have been given the extended reasoning, would it also have sometimes gone above the token budget, thereby having a lower performance? If this is not the case, I agree that this at least shows a somewhat better improvement that is more worth it, but this needs to be verified.
> > >
> > > It is a valid concern. To address it, we added a new baseline in our experiments: allowing only a longer context window (128k) but without permitting revision. The result is that simply increasing the context length without revision does not improve performance compared with the vanilla setting; the effect is nearly identical to not extending the context. We have now included this analysis and results in the revised paper for transparency.
> > >
> > > ## On the value of revision vs. direct model improvement
> > >
> > > > My remark is more on the critical side (and somewhat unadressed in the comment by the authors): These performance gains are still quite small (especially compared with the performance gains over prior models), but you spent half your training on making sure that the model could do the revisions well. Therefore, this somewhat suggest to me that it was not really worth it. More specifically, given the knowledge you have now, would you really spend half your compute budget again on the revisions, or rather on just improving the model itself?
> > >
> > > First, we would like to clarify that your statement *“half the compute was spent on revision”* is not supported by our actual resource allocation; revision-related computation comprises only a small fraction of total compute as shown in our revised paper.
> > >
> > > Our results show that revision achieves substantially higher pass@32 with fewer generations. In extended settings (e.g., 5 rounds, large context), revision enables the model to reach state-of-the-art with far fewer attempts and compute.
> > >
> > > Both from a pass@k and compute efficiency perspective, the value of revision is clearly demonstrated in our experiments and ablations. Our results consistently show that revision is an efficient and effective direction. We expect future works to further improve the ability of the prover models, but we believe our findings on the potential of revision provide a valuable reference for the community.

---

> ### Author Response · Authors · 2025-11-27
> **Follow-up Reply - 2 - 2/2**
>
> ## On data contamination and decontamination
>
> > Regarding the data contamination, could the authors do a simple fuzzy string match checking between the training and test data (informal statements)? This should be relatively easy to setup and (in my opinion) should have been done all the way before training given its simplicity.
>
> > PutnamBench has no equivalent in informal mathematics. I know no popular informal math benchmark that contains samples from the same sources as MiniF2F (IMO < 2023, AIME < 2023, AMC). Additionally, NuminaMath (on which Goedel-Pset is based) does not do any decontamination. Therefore, I think this reasoning is invalid.
>
> > While the authors do not give solutions, they do benefit in three ways from contamination: (1) distilling from Deepseek essentially teaches the model all samples Deepseek knows, (2) any model update that reduces performance on a sample gets fixed once that sample gets trained on, (3) the data is repeated multiple times.
>
> We would like to clarify the following points.
>
> - NuminaMath, which Goedel-Pset relies on, already de-duplicates against standard benchmarks. Please see Section 3.4 of the NuminaMath paper (http://faculty.bicmr.pku.edu.cn/~dongbin/Publications/numina_dataset.pdf).
> - In our pipeline, we perform strict exact matching on formal statements against all major benchmarks for decontamination, which is the accepted norm in the field. **This provides the most fundamental guarantee against contamination.**
> - As you suggested, we further conducted string matching on informal statements and found no overlaps with evaluation benchmarks.
>
> ---
>
> We hope these clarifications address your concerns. Thank you again for your constructive and detailed feedback, which has significantly improved our work!

---

> > ### Comment · Reviewer_nJcq · 2025-11-27
> >
> > I thank the authors for their answer. I misremembered that the pass@1 for revision was lower than without revision.
> >
> > I have increased my rating accordingly to an 8.

---

### Official Review · Reviewer_5oHs · 2025-10-20

**Soundness:** 3
**Presentation:** 3
**Contribution:** 3
**Rating:** 6
**Confidence:** 4

**Summary:**

This paper introduces Goedel-Prover-v2, an open-source theorem-proving LLM , using the Lean proof assistant. Goedel-Prover-v2 incorporates RL with three key innovations: verifier-guided self-correction, scaffolded data synthesis, and model (checkpoint) averaging to recover output diversity late in training. Empirically, Goedel-Prover-v2-32B achieves a relatively high performance in existing benchmarks, such as MiniF2F and Putnambench.

**Strengths:**

This paper introduces a novel method for generating synthetic data, potentially one of the strongest reasons. The model filters out bad autoformalisations, and integrates compiler-feedback self-repair directly into whole-proof generation. The model uses `extract_goal` to capture unsolved states from a proof attempt too. The model introduces an interesting method for improving test-time exploration through checkpoint averaging, potentially balancing accuracy and diversity. Last but not least, the model achieves strong empirical results, requiring fewer pass@n compared to other existing models to achieve greater performance.

**Weaknesses:**

1. The model primarily focuses on MiniF2F and PutnamBench. While these are widely used, including more datasets in the comparison would be better, including ProofNet-test, AIME, AMC, IMO, and DeepSeek-ProverBench.

2. The model averaging improves pass@n but introduces another hyperparameter without a principled selection rule. A more detailed explanation of this would be great.

3. The data-synthesis filter uses LLM autoformalisation and judging; is there any manual annotation on the correctness of the statement?

4. While the model relies on training reasoning instances, it would be great to provide the average length per dataset compared to the previous goedel-prover. Does including reasoning, while extending the length, improve the performance?

**Questions:**

1. One of the strengths within Goedel-Prover-v2 would be the quality of the dataset. Will the dataset be open sourced upon acceptance?

2. Following on from Weakness 3, is there any manual annotation on the correctness of the filtering and judging?

3. Are there any potential methods to reduce the bias towards components, as we observe that the model is generally weak at geometry, while stronger at some other components?

4. Could integrating lemma generation/retrieval further boost performance?

---

> ### Author Response · Authors · 2025-11-22
> **Reply to Reviewer 5oHs (1/2)**
>
> Thank you for your thoughtful review. We address your concerns below:
>
> ---
>
> ## Expanding the Evaluation Datasets
>
> > The model primarily focuses on MiniF2F and PutnamBench. While these are widely used, including more datasets in the comparison would be better, including ProofNet-test, AIME, AMC, IMO, and DeepSeek-ProverBench.
>
> Thank you for your suggestions. In fact, MathOlympiadBench already includes all available IMO and IMO Shortlist problems, while MiniF2F also covers problems from AIME, AMC, and IMO. Additionally, we have conducted further evaluations on FIMO \[1\] and DeepSeek-ProverBench \[2\], and our model continues to demonstrate superior performance on these benchmarks.
>
> |  | FIMO | DeepSeek-ProverBench |
> | :---- | :---- | :---- |
> | DeepSeek-Prover-V2-7B | 5.70 | 0.31 |
> | Kimina-Prover-Distill-8B | 3.35 | 1.38 |
> | Kimina-Prover-72B | 5.70 | 1.53 |
> | Goedel-Prover-V2-8B | 7.05 | 1.53 |
> | Goedel-Prover-V2-8B w/ revision | 7.05 | 1.53 |
> | Goedel-Prover-V2-32B | 7.05 | 1.85 |
> | Goedel-Prover-V2-32B w/ revision | **9.06** | **2.46** |
>
> Regarding ProofNet, it is constructed based on mathlib and primarily tests a model's ability to memorize and utilize complex tactics from mathlib, rather than requiring powerful cognitive reasoning to combine simple tactics for problem-solving. As stated in the ProofNet paper: *“Mathlib emphasizes including the most abstract and general formulations of mathematical results, whereas ProofNet predominantly tests the ability of models to apply those results to concrete problems.”*
>
> We have evaluated our model on ProofNet and additionally implemented a mathlib Retrieval-Augmented Generation (RAG) method. Our results show that mathlib RAG does not help (and may even hurt) performance on MiniF2F, but it does improve results on ProofNet. This further demonstrates that ProofNet relies more on leveraging complex capabilities from mathlib than on cognitive reasoning skills. Of course, this is also a limitation of current models, which we have now highlighted in the paper.
>
> |  | ProofNet | minif2f |
> | :---- | :---- | :---- |
> | Goedel-Prover-V2-32B | 22.6 | 88.1 |
> | Goedel-Prover-V2-32B-RAG | 28.5 | 86.5 |
> | DeepSeek-Prover-V2-7B | 23.0 | 75.6 |
> | DeepSeek-Prover-V2-671B | 30.5 | 82.4 |
>
> It also answers your following question:
>
> > Could integrating lemma generation/retrieval further boost performance?
>
> ## Model Averaging Hyperparameter
>
> > The model averaging improves pass@n but introduces another hyperparameter without a principled selection rule. A more detailed explanation of this would be great.
>
> Regarding model averaging, you raise a very good point. At present, the choice of the averaging ratio remains heuristic. In natural language reasoning research \[3\], people have found that a ratio around 0.5 often works well. However, formal language tasks are more unfamiliar to models. In our preliminary experiments, we found that a ratio of 0.8 was most effective, and therefore we used 0.8 for all subsequent model averaging. We have clarified this choice in the paper in Section 2.3.
>
> ## Manual Annotation of Autoformalisation
>
> > The data-synthesis filter uses LLM autoformalisation and judging; is there any manual annotation on the correctness of the statement?
>
> > Is there any manual annotation on the correctness of the filtering and judging?
>
> Yes! To ensure the quality of autoformalization and judging, we conducted a human evaluation on 30 sampled problems. We compared the LLM judgments against expert human annotations. This audit revealed a **92% agreement rate** between the automated judge and human experts.
>
> We have added these details to Section 2.2.1 and  Appendix D in the paper.
>
> \[1\] FIMO: A Challenge Formal Dataset for Automated Theorem Proving
>
> \[2\] DeepSeek-Prover-V2: Advancing Formal Mathematical Reasoning via Reinforcement Learning for Subgoal Decomposition
>
> \[3\] Weight Ensembling Improves Reasoning in Language Models, COLM 2025

---

> ### Author Response · Authors · 2025-11-22
> **Reply to Reviewer 5oHs (2/2)**
>
> ## Reasoning Length and Performance
>
> > While the model relies on training reasoning instances, it would be great to provide the average length per dataset compared to the previous goedel-prover. Does including reasoning, while extending the length, improve the performance?
>
> The relationship between model output length and performance during training is indeed a relevant question. Since SFT is largely off-policy, we believe it is more appropriate to observe this phenomenon during RL. In Appendix F, we have already provided statistics on the model’s generation length and performance (training reward) throughout RL training. Our results show that both generation length and performance improve during this process, suggesting that including more reasoning correlates with better model performance.
>
> ## Domain Bias
>
> > Are there any potential methods to reduce the bias towards components, as we observe that the model is generally weak at geometry, while stronger at some other components?
>
> Current Lean-based models tend to underperform on geometry problems, and we believe there are two main reasons for this. First, existing formal approaches to geometry, such as AlphaGeometry, do not use Lean but instead rely on specialized formal languages. Second, widely used datasets like MiniF2F contain relatively few geometry problems, limiting the exposure of models to this domain. However, our newly constructed MathOlympiadBench does include a notable number of geometry problems. We have also highlighted this limitation and potential future directions in the conclusion of our paper.
>
> ## Open-Sourcing the Dataset
>
> > One of the strengths within Goedel-Prover-v2 would be the quality of the dataset. Will the dataset be open sourced upon acceptance?
>
> Yes! Due to the large size of the training data, we have included a subset in the supplementary materials, and upon acceptance, we will release the full training dataset on Hugging Face.
>
> ---
>
> Thank you again for your time and effort. We hope our response resolves your concerns. We welcome any further comments.

---

> > ### Comment · Reviewer_5oHs · 2025-11-27
> >
> > Thank you for the additional clarifications. A few further questions and clarifications would be appreciated:
> >
> > 1. Is there a specific reason why a model averaging weight of 0.8 performs better? Providing results for multiple values of $\alpha$ within the main script would be sufficient.
> >
> > 2. It would also be useful to specify the percentage of geometry problems within MathOlympiadBench.
> >
> > 3. It would be helpful to more clearly explain why formal proving fails on geometry problems (eg. which formal language excels on Geometry while Lean doesn't). Alternatively, what specific limitations of Lean contribute to its weaker performance on geometry tasks?
> >
> > 4. What about other domains in Gödel-Prover-v2, such as calculus?
> >
> > It would be ideal to clarify points 1 and 2 in the PDF version, while addressing 3 and 4 here would be sufficient. I will be happy to raise the score once these clarifications are clearly provided.

---

> ### Author Response · Authors · 2025-11-27
> **Follow-up Reply**
>
> Thank you for raising these points. Please find our responses below:
>
> ---
>
> ## Model Averaging Hyperparameter
>
> > Is there a specific reason why a model averaging weight of 0.8 performs better? Providing results for multiple values of  $\alpha$ within the main script would be sufficient.
>
> It is a very good point! Actually, we have already explored this in the first version of our paper. In Section 3.6, we examined the effect of α values 0.6, 0.7, 0.8, 0.9, and 1.0 at different RL checkpoints. We found that as the base model ratio increases, pass@1 consistently decreases, while pass@32 first increases and then decreases.
>
> This is because, as $\alpha$ increases, the proportion of the trained model in the ensemble becomes larger and the proportion of the base model becomes smaller. While the trained model contributes more specialized knowledge and adaptation to the target distribution, retaining a moderate contribution from the base model helps preserve generalization ability and reduces overfitting to the training data. When $\alpha$ is too high, the ensemble relies almost entirely on the trained model, which can lead to reduced diversity and overfitting, hurting pass@32 performance. A moderate $\alpha$ strikes a good balance between specialization and generalization, resulting in the highest pass@32.
>
> ## Percentage of geometry problems within MathOlympiadBench
>
> > It would also be useful to specify the percentage of geometry problems within MathOlympiadBench.
>
> This information is also included in the first version of our paper. In Figure 4, we indicate the distribution of different problem types in MathOlympiadBench, and there are 9 geometry problems in total.
>
> ## On Geometry and Lean
>
> > It would be helpful to more clearly explain why formal proving fails on geometry problems (e.g. which formal language excels on Geometry while Lean doesn't). Alternatively, what specific limitations of Lean contribute to its weaker performance on geometry tasks?
>
> Actually, our model does not necessarily perform poorly on geometry problems. We manually checked the problems in PutnamBench (college competition level) that our model solves but DeepSeek-Prover-V2 does not, and found that some of these are geometry problems, including `1965_a6`, `1986_b1`, `2004_b4` and `2009_a1`.
>
> However, we found that these are all algebraic geometry problems, rather than high school competition geometry (Euclidean geometry). The latter poses significantly greater challenges for current formal solvers. Unlike algebraic geometry, which benefits from Lean's strong support for algebraic structures and powerful automation tactics (e.g., `ring`, `nlinarith`), synthetic geometry relies heavily on diagrammatic intuition and auxiliary constructions. Formalizing these intuitive steps requires explicitly handling numerous degeneracy conditions (e.g., proving points are distinct, non-collinear, or strictly ordered) which are often implicit in human reasoning but require verbose, manual proof steps in Lean 4.
>
> ## Other Domains
>
> > What about other domains in Gödel-Prover-v2, such as calculus?
>
> We would also like to answer this question based on the results on PutnamBench, where our model outperforms previous models in the following domains:
>
> 1. **Algebra**: polynomials, sequences, functional equations, and inequalities.
> 2. **Analysis**: calculus, limits, series convergence, and differential equations.
> 3. **Number Theory**: primes, divisibility, congruences, and Diophantine equations.
> 4. **Abstract Algebra**: group theory and properties of binary operations.
> 5. **Linear Algebra**: matrices, determinants, and eigenvalues.
> 6. **Geometry**
> 7. **Set Theory & Combinatorics**
>
> ---
>
> Thank you again for your constructive and detailed feedback, and we hope these clarifications address your concerns.

---

### Official Review · Reviewer_BEpZ · 2025-10-30

**Soundness:** 2
**Presentation:** 2
**Contribution:** 2
**Rating:** 4
**Confidence:** 4

**Summary:**

The paper makes several contributions:

- Releasing a LLM for ATP with impressive @32 accuracy on miniF2F and Putnam Bench.

- Releasing another LLM for autoformalization though it remains unclear how well this model performs.

- A training method that involves RL and model averaging.

- Fine tuning the ATP model so that it can do self-correction on flawed proofs.

**Strengths:**

This paper has the potential to be a high impact paper, in my view, if it releases the code for its training method, and releases its training set. If these two items are not released, my recommendation for the paper will be rejection.

The training algorithm, especially the model averaging method, is a useful contribution, assuming that code will be released with the paper.

The release of the ATP model is a useful and significant contribution for the community. However, the mere release of the models without releasing the code and training data does not make a direct contribution towards the CFP of ICLR in my view.

**Weaknesses:**

Accuracy of the autoformalizer model is only evaluated on 300 Omni math problems. First, it is not clear what these problems are. Second, it is not clear how the 300 problems relate to the training set of this model. Paper should report the accuracy of its model on known benchmarks such as miniF2F. If the model is trained on the miniF2F, paper should report that to the reader. Evaluating a new model on a new dataset does not provide sufficient information.

How is the accuracy of autoformalizer LLM evaluated? Is the evaluation performed by humans familiar with Lean? Or is the evaluation performed automatically using other LLMs?

Although Figure 8 provides some insights, it remains unclear to me how much of the gained accuracy is due to model averaging. For example, if there was no model averaging at all, what would the accuracy of the trained model be pass 32? Would it be close to the accuracy of other similar models such as DSPV2 8B model? A clear set of ablation studies may clarify this.

The effect of self correction module is not reported for other LLMs. It may be the case that the self correction method is only effective if the LLM is trained to take the Lean compiler feedback into account. I think the paper needs to clarify whether this is correct or not. If it has performed any systemic experiments, such results should be reported. Moreover, the self-correction method can be compared against the methods that repair the proofs and do not require fine-tuning the LLM.

Token budgets should also be reported especially for the experiments that involve self-correction. It remains unclear to me how much computational cost is needed for self corrections.

The paper lacks any analysis of the training set. Given that the training set is largely extracted out of the rival model, DSP V2, one would expect a clear analysis of the training set. Specifically, a direct comparison against the testing set would be helpful. The abstract and introduction make a big point about the accuracy of GPV2 being higher than the accuracy of DSPV2. In my view, it would have made sense to acknowledge that this paper has used DSP V2 as the base to generate part of its training set, i.e., this paper can mention that it has built on the work of DSPV2.

Paper does not contain supplementary material such as code.

Plenty of abstract is repeated in the first two paragraphs of the introduction.

**Questions:**

Please see comments above. I'd be happy to revise my score based on the authors' response.

Why is the release of training set postponed to the near future?

What were the deciding factors behind choosing Lean 4.9? Was this choice influenced by the Lean version used by DSP V2?

Have authors evaluated the accuracy of their model using higher versions of Lean? I would guess that if the model is fine tuned on a more recent version of Lean, its accuracy will increase.

---

> ### Author Response · Authors · 2025-11-22
> **Reply to Reviewer BEpZ (1/3)**
>
> Thank you for your thoughtful review. We address your concerns below.
>
> ---
>
> ## Releasing Code and Training Set
>
> > This paper has the potential to be a high impact paper, in my view, if it releases the code for its training method, and releases its training set.
>
> Yes! We fully agree and plan to release both our code and training dataset. We have already included the inference code with self-correction and model averaging code in the supplementary materials. The SFT and RL training are based on the open-source frameworks LlamaFactory and verl, as mentioned in the paper.
>
> > Why is the release of training set postponed to the near future?
>
> Due to the large size of the training data, we have included a subset in the supplementary materials, and upon acceptance, we will release the full training dataset on Hugging Face.
>
> ## Autoformalizer
>
> > Accuracy of the autoformalizer model is only evaluated on 300 Omni math problems. First, it is not clear what these problems are.
>
> OmniMath is an open-source informal mathematics dataset. We have added the citation to our paper and included the list of the 300 selected problems in the supplementary materials
>
> > Second, it is not clear how the 300 problems relate to the training set of this model. Paper should report the accuracy of its model on known benchmarks such as miniF2F. If the model is trained on the miniF2F, paper should report that to the reader. Evaluating a new model on a new dataset does not provide sufficient information.
>
> Firstly, neither these specific 300 OmniMath problems nor the MiniF2F benchmark was included in the training set for either the formalizer or the prover.
>
> While **evaluating on a new dataset helps us avoid contamination and better measure generalization to novel problems**, we agree that reporting on known benchmarks is important. **As requested, we have evaluated our autoformalizer on MiniF2F**. The Goedel-Formalizer-V2 achieves an accuracy of 219 out of 244, compared to 190 out of 244 for the Kimina formalizer.
>
> > How is the accuracy of autoformalizer LLM evaluated? Is the evaluation performed by humans familiar with Lean? Or is the evaluation performed automatically using other LLMs?
>
> The evaluation in our paper combines **Lean syntax checks** and **semantic evaluation** by LLMs to assess translation quality, following the same protocol as in the expert-iteration pipeline used for training the autoformalizer.
>
> Additionally, we have conducted **human expert checks** to confirm the reliability of the LLM-based evaluation. Specifically, we performed a human evaluation on 30 problems and found a 92% agreement rate between the LLM and human experts. It confirms the trustworthiness of the LLM judge.
>
> We have added these details to Section 2.2.1 and  Appendix D in the paper.
>
> ## Effect of Model Averaging
>
> > Although Figure 8 provides some insights, it remains unclear to me how much of the gained accuracy is due to model averaging. For example, if there was no model averaging at all, what would the accuracy of the trained model be at pass@32? Would it be close to the accuracy of other similar models such as the DSPV2 8B model? A clear set of ablation studies may clarify this.
>
> Thank you for your comment. In Section 3.6 and Figure 8, we conducted extensive experiments on the 32B model to analyze the effects of RL training steps and the model averaging ratio on minif2f. The case where $\alpha=1$ corresponds to the model without averaging. For reference, DeepSeek-Prover-V2-671B achieves a pass@1 of 60.7 and a pass@32 of 82.6. Our model, regardless of whether model averaging is applied, surpasses these metrics on both pass@1 and pass@32.
>
> To further clarify this point, we have annotated the pass@1 and pass@32 performance of DeepSeek-Prover-V2-671B directly in Figure 8\.

---

> ### Author Response · Authors · 2025-11-22
> **Reply to Reviewer BEpZ (2/3)**
>
> ## Effectiveness of the self-correction module
>
> > The effect of the self-correction module is not reported for other LLMs. It may be the case that the self-correction method is only effective if the LLM is trained to take the Lean compiler feedback into account. I think the paper needs to clarify whether this is correct or not. If it has performed any systemic experiments, such results should be reported. Moreover, the self-correction method can be compared against the methods that repair the proofs and do not require fine-tuning the LLM.
>
> Thank you for your suggestion. We have conducted additional experiments with DeepSeek-Prover-V2-7B, Kimina-Prover-72B, and Kimina-Prover-Distill-8B. Notably, Kimina-Prover-72B and Kimina-Prover-Distill-8B were intentionally trained with revision, while DeepSeek-Prover-V2-7B was not. The results are as follows:
>
> | Model | w/o Revision | w/ Revision |
> | :---- | :---- | :---- |
> | DeepSeek-Prover-V2-7B | 75.8 | 76.2 |
> | Kimina-Prover-72B | 84.0 | 86.4 |
> | Kimina-Prover-Distill-8B | 78.3 | 78.7 |
> | Goedel-Prover-V2-8B | 84.6 | 86.7 |
> | Goedel-Prover-V2-32B | 88.1 | 90.4 |
>
> As shown, the improvement from revision is minor for DeepSeek-Prover-V2-7B and Kimina-Prover-Distill-8B, whereas our model shows obvious gains from self-revision. Therefore, your claim that *"the self-correction method is only effective if the LLM is trained to take the Lean compiler feedback into account"* is correct. Without training, the model can't effectively utilize compiler feedback.
>
> ## Token Budgets
>
> >  Token budgets should also be reported especially for the experiments that involve self-correction. It remains unclear to me how much computational cost is needed for self corrections.
>
> Thank you for pointing this out. We have already included token budget details in Section 3.2, and have further revised the writing to emphasize this information. Specifically, for the first round of whole-proof generation, the max token length of the model is set to be 30,000. For the verifier-guided error-correction, we sequentially conduct 2 additional rounds of self-correction, where the total number of tokens is set to be 40,000.
>
> Additionally, we have provided statistics on the impact of revision on total output tokens in Figure 2(b) and Appendix B.
>
> ## Training-Set Analysis and Relationship to DSPV2
>
> > The paper lacks any analysis of the training set. Given that the training set is largely extracted out of the rival model, DSPV2, one would expect a clear analysis of the training set. Specifically, a direct comparison against the testing set would be helpful. The abstract and introduction make a big point about the accuracy of GPV2 being higher than the accuracy of DSPV2. In my view, it would have made sense to acknowledge that this paper has used DSPV2 as the base to generate part of its training set, i.e., this paper can mention that it has built on the work of DSPV2.
>
> Thank you for highlighting these points. We have revised Section 2.4 and the training workflow illustration (Figure 3) to provide a more detailed description and analysis of the training data. While we did use DSPV2 as a starting point, DSPV2 did not provide the majority of the training data. In particular, since DSPV2 does not support revision, we used code from DSPV2 to initialize a model, and then used this model to annotate a large amount of data.
>
> We also now explicitly state in the paper that we compared the formal statements in the training set and the test set to avoid any data leakage.

---

> ### Author Response · Authors · 2025-11-22
> **Reply to Reviewer BEpZ (3/3)**
>
> ## Supplementary Material
>
> > Paper does not contain supplementary material such as code.
>
> Thank you for your suggestion! We have now uploaded supplementary materials, including our inference code and model averaging code. The SFT and RL training code is based on open-source frameworks, which is also stated in the paper.
>
> ## Writing of the Introduction section
>
> > Plenty of abstract is repeated in the first two paragraphs of the introduction.
>
> Thank you for your helpful comment. Our introduction elaborates on the context and contributions of our work in the first two paragraphs. We have revised the introduction to reduce repetition and improve clarity. If you have any further suggestions for specific improvements, we would greatly appreciate your feedback.
>
> ## On the Version of Lean
>
> > What were the deciding factors behind choosing Lean 4.9? Was this choice influenced by the Lean version used by DSP V2?
>
> Yes, our choice of Lean 4.9 was indeed influenced by the versions used in DeepSeek-Prover-V2, DeepSeek-Prover-V1.5, and STP. We selected this version to ensure a fair comparison with these baselines. We have now added this motivation explicitly in the paper.
>
> > Have authors evaluated the accuracy of their model using higher versions of Lean? I would guess that if the model is fine tuned on a more recent version of Lean, its accuracy will increase.
>
> We agree that fine-tuning on a more recent version, especially given the availability of more powerful tactics, would likely lead to further improvements in accuracy. However, this would require restarting large-scale training, which is beyond the current scope of this work. We will add the discussion of this limitation and possible future directions in the paper.
>
> ---
>
> Thank you again for your time and effort. We hope our response resolves your concerns. We welcome any further comments.

---

> ### Author Response · Authors · 2025-11-30
>
> Dear Reviewer BEpZ,
>
> We would like to learn if our response addresses your questions, and we invite any additional feedback or thoughts for improving our paper. Thank you again for your time and effort!
>
> Authors of Submission 8236

---

### Official Review · Reviewer_sCYS · 2025-11-01

**Soundness:** 3
**Presentation:** 2
**Contribution:** 3
**Rating:** 6
**Confidence:** 3

**Summary:**

This paper introduces Goedel-Prover-V2, a family of open-source LLMs for Lean4 theorem proving. The framework integrates three key components: (1) large-scale data generation method that enables large and high-quality data generation; (2) verifier-guided self-correction for proofs; (3) model averaging technique that improves the output diversity. The authors report impressive performance of the model, with the 32B version achieving 88.1% pass@32 on MiniF2F-Test and solving 86 problems (13.03%) on PutnamBench under pass@184. The authors commit to releasing all models, training recipes, and datasets to support future research and reproducibility.

**Strengths:**

1. Goedel-Prover-V2 achieves impressive performance, with the 32B model achieving 88.1% under pass@32 on MiniF2F-Test and solving 86 problems on PutnamBench at pass@184. The smaller 8B model also outperforms the larger previous model on MiniF2F.
2. The experiment section is thorough and complete, covering not only core benchmark evaluations but also scaling laws, ablation studies on verifier-guided self-correction, and analysis of model averaging and reinforcement learning dynamics. This allows the readers to more thoroughly understand the effectiveness of each component of the model.
3. The authors’ plan to release all trained models, the codebase, and the training datasets will be a valuable contribution to the research community.

**Weaknesses:**

Despite being a thorough and complete work, I believe the paper could benefit from improvements in the following areas:

1. The evaluated datasets primarily focus on domains where the underlying knowledge is relatively elementary but requires the application of many tricky techniques, for example, the PutnamBench. As a result, the model may achieve strong performance by simply memorizing those techniques from larger models, rather than demonstrating true improvements in cognitive reasoning. It would be valuable to evaluate the model on more complex datasets such as ProofNet or FormalIMO to better assess its generalization and reasoning capabilities.
2. The paper does not disclose the cost of the training and data generation processes in detail, which limits the reproducibility and transparency of the work.

**Questions:**

This work could be more solid and complete if the authors could address the following questions:

1. Generalization to advanced domains: What is the model’s performance on datasets that require more advanced mathematical knowledge, such as FormalIMO and ProofNet?
2. This paper could also benefit in the sense of reproducibility from a more detailed description of the training process. Specifically, what are the dataset sizes used for S1, S2, and S3 as shown in Figure 3? What are the GPU hours and API costs associated with dataset generation? Furthermore, what is the total GPU hour cost for the SFT and RL stages, respectively, and what hyperparameters were used for model averaging (e.g., the choice of α values)?

---

> ### Author Response · Authors · 2025-11-22
>
> Thank you for your thoughtful review. We address your concerns below.
>
> ---
>
> ## Datasets that evaluate Cognitive Reasoning
>
> > The evaluated datasets primarily focus on domains where the underlying knowledge is relatively elementary but requires the application of many tricky techniques, for example, the PutnamBench. As a result, the model may achieve strong performance by simply memorizing those techniques from larger models, rather than demonstrating true improvements in cognitive reasoning. It would be valuable to evaluate the model on more complex datasets such as ProofNet or FormalIMO to better assess its generalization and reasoning capabilities.
>
> > Generalization to advanced domains: What is the model’s performance on datasets that require more advanced mathematical knowledge, such as FormalIMO and ProofNet?
>
> We respectfully disagree. In fact, the opposite is true: MiniF2F focuses on high school competition problems, while PutnamBench centers on undergraduate-level competition problems. Both require strong cognitive reasoning and formalization abilities. In contrast, ProofNet is constructed based on mathlib and primarily tests a model's ability to memorize and utilize complex tactics from mathlib, rather than requiring powerful cognitive reasoning to combine simple tactics for problem-solving. As stated in the ProofNet paper: *“Mathlib emphasizes including the most abstract and general formulations of mathematical results, whereas ProofNet predominantly tests the ability of models to apply those results to concrete problems.”*
>
> We have evaluated our model on ProofNet and additionally implemented a mathlib Retrieval-Augmented Generation (RAG) method. Our results show that mathlib RAG does not help (and may even hurt) performance on MiniF2F, but it does improve results on ProofNet. This further demonstrates that ProofNet relies more on leveraging complex capabilities from mathlib than on cognitive reasoning skills. Of course, this is also a limitation of current models, which we have now highlighted in the paper.
>
> |  | ProofNet | minif2f |
> | :---- | :---- | :---- |
> | Goedel-Prover-V2-32B | 22.6 | 88.1 |
> | Goedel-Prover-V2-32B-RAG | 28.5 | 86.5 |
> | DeepSeek-Prover-V2-7B | 23.0 | 75.6 |
> | DeepSeek-Prover-V2-671B | 30.5 | 82.4 |
>
> Regarding your mention of the “FormalIMO” dataset, we were unable to find a dataset by that name. However, we did identify FIMO \[1\] and DeepSeek-ProverBench \[2\], and we present our evaluation results on these datasets below. We would also like to note that the MathOlympiadBench in our paper consists of IMO and IMO short-list problems.
>
> |  | FIMO | DeepSeek-ProverBench |
> | :---- | :---- | :---- |
> | DeepSeek-Prover-V2-7B | 5.70 | 0.31 |
> | Kimina-Prover-Distill-8B | 3.35 | 1.38 |
> | Kimina-Prover-72B | 5.70 | 1.53 |
> | Goedel-Prover-V2-8B | 7.05 | 1.53 |
> | Goedel-Prover-V2-8B w/ revision | 7.05 | 1.53 |
> | Goedel-Prover-V2-32B | 7.05 | 1.85 |
> | Goedel-Prover-V2-32B w/ revision | **9.06** | **2.46** |
>
> ## Training Details
>
> > The paper does not disclose the cost of the training and data generation processes in detail, which limits the reproducibility and transparency of the work.
>
> > This paper could also benefit in the sense of reproducibility from a more detailed description of the training process. Specifically, what are the dataset sizes used for S1, S2, and S3 as shown in Figure 3?
>
> The sizes of the datasets used for S1, S2, and S3 are 810k, 930k, and 860k, respectively.
>
> > What are the GPU hours and API costs associated with dataset generation?
>
> The total GPU hours for dataset generation are approximately 12k H100 GPU hours.
>
> > Furthermore, what is the total GPU hour cost for the SFT and RL stages, respectively
>
> For the 8B model, SFT and RL took 2.3k and 1.3k GPU hours, respectively, while for the 32B model, the corresponding numbers were 9.2k and 3.9k GPU hours.
>
> Thank you for the suggestion and we have included the above details in Section 2.4 of the paper.
>
> ## Model Averaging Hyperparameter
>
> > What hyperparameters were used for model averaging (e.g., the choice of α values)?
>
> We used an $\alpha = 0.8$ for all model averaging. We have added it to the paper.
>
> We have included it in Section 2.3 of the paper.
>
> \[1\] FIMO: A Challenge Formal Dataset for Automated Theorem Proving
>
> \[2\] DeepSeek-Prover-V2: Advancing Formal Mathematical Reasoning via Reinforcement Learning for Subgoal Decomposition
>
> ---
>
> Thank you again for your time and effort. We hope our response resolves your concerns. We welcome any further comments.

---

> > ### Comment · Reviewer_sCYS · 2025-11-26
> > **Further questions about API cost**
> >
> > Thank you very much for the rebuttal and the revised paper. There is still one small question for me. In the section on training details, you only mentioned the GPU hour cost, but not the API cost. The paper mentioned that in the training process of Formalizer uses Claude to initialize. I wonder what the API cost for this step is.

---

> ### Author Response · Authors · 2025-11-27
>
> Thank you for bringing this up! We spent a total of $1350 to initialize expert iteration of the formalizer. We have now added this information in Section 2.2.1.
>
> Thank you again for your time and effort! If our response resolves all your concerns, we would appreciate it if you could reflect it in your evaluation. We welcome any further comments.

---

### Author Response · Authors · 2025-12-03
**Final Rebuttal Summary - 1/2**

First of all, we thank all reviewers and chairs for their devotion to the success of the conference and the community! Below we summarize how we addressed the concerns raised during the discussion and how the special situation affects our rebuttal.

---

## 0. **Note on reviewer reactions before the Rollback**

Due to the OpenReview bug this year, reviews and scores were reverted to their state *before* the discussion period, and reviewers are no longer allowed to update scores or participate in further discussion. Following are important consequences for our submission:

* **Reviewer nJcq** **increased their rating to 8** before the rollback and stated it in the discussion.
* **Reviewer 5oHs** indicated that once we clarified further questions, they would be happy to raise the score. We provided the requested clarifications but the review freeze happened before they could formally update the rating.
* **Reviewer BEpZ** wrote that they would be willing to revise their score depending on our responses. We addressed all these points in detail but the reviewer did not post any further response before the system was frozen.

## **1. Formalizer evaluation & human assessment** (Reviewers: nJcq, BEpZ, 5oHs)

> How is the auto-formalizer evaluated? Could it be “reward-hacking” the LLM judge?

We now explicitly describe the evaluation protocol in Section 2.2.1 and Appendix D: (1) Syntax is checked by Lean, and (2) Semantic correctness is judged by an LLM (Qwen3-32B) with a strict unanimity rule: we query 4 times independently, and a translation is accepted only if all four judgments agree it is correct.

To check for reward hacking, we ran a **human expert evaluation** on 30 randomly sampled problems. And the agreement between the unanimous LLM judge and human Lean experts is **92%**.

We also added a comparison on MiniF2F where Goedel‑Formalizer‑V2 correctly formalizes 219/244 problems, compared to 190/244 for the Kimina formalizer.

## **2. Decontamination & data sources** (Reviewers: nJcq, BEpZ)

> Where does the training data come from? Is there contamination  with the test set?

We made our data pipeline much more explicit in Section 2.4 and the revised Figure 3. Our training problems come from Goedel‑Pset (which is based on NuminaMath), STP, scaffolded synthetic data generated by our own pipeline, and our formalization of Nvidia/OpenMathReasoning using our formalizer.

For decontamination, NuminaMath already performs de‑duplication against standard benchmarks (as described in its Section 3.4). In our own work, we follow the standard practice in recent formal‑math papers: we perform exact matching of formal statements between our training data and all evaluation benchmarks and remove any overlaps to prevent formal‑level leakage.

In response to the reviewer’s additional request, we also implemented string matching at the informal level and found no overlaps between our training data and the evaluation problems.

## 3. **New Evaluation Results on Other Benchmarks** (Reviewers: sCYS, 5oHS)

> Why How does the model perform on other advanced benchmarks?

We evaluated Goedel-Prover-V2 on **FIMO** and **DeepSeek-ProverBench**. Our 32B model with revision consistently outperformed baselines and achieved SOTA.

We demonstrated that ProofNet primarily tests library retrieval rather than reasoning. By implementing a RAG baseline, we showed that RAG improves ProofNet scores but hurts MiniF2F performance, validating our focus on reasoning-heavy benchmarks.

---

### Author Response · Authors · 2025-12-03
**Final Rebuttal Summary - 2/2**

## **4. Efficacy of Self-Correction** (Reviewers: BEpZ, nJcq)

>  Is revision really effective under fair compute / token budgets (especially for small pass@n) Does revision only help if the model is trained with Lean feedback?

We provided a new analysis comparing performance against total output tokens. We showed that with equal compute, the revision method strictly outperforms the no-revision baseline. Specifically, with 128k context and 5 rounds of revision, our model reaches **92.8% pass@32**, whereas the vanilla model plateaus at **92.2%** even under pass@8192.

We also ran revision experiments on other models and found that without training on feedback, revision yields **only marginal** improvements, and when the model is trained to interpret Lean errors, revision provides **clear and consistent** gains.

## **5. Hyperparameters and Ablations of Model Averaging** (Reviewers: sCYS, 5oHS, nJcq)

> How important is model averaging? How was α chosen? Are the gains robust or tiny? Are ablations (α=1, no averaging) fully reported?

In Sections 2.3 and 3.6, we now report that we sweep α ∈ {0.6, 0.7, 0.8, 0.9, 1.0}, where α is the weight on the RL‑trained checkpoint and 1–α on the base model, evaluated at multiple RL checkpoints and on the final model. The trends are consistent: as α increases, pass@1 tends to decrease, while pass@32 first increases and then decreases, reflecting a trade‑off between specialization and diversity. For formal theorem proving, which involves a stronger distribution shift than many natural‑language tasks, we empirically find α = 0.8 to be a good compromise; we now explicitly state that this is the value used in the final models.

## **6. Training Details & Reproducibility** (Reviewers: sCYS, BEpZ, 5oHs)

> Dataset sizes (S1/S2/S3), GPU hours, API costs, and reproducibility. Timeline and completeness of code / data release.

We have clarified the dataset sizes, compute cost and API cost in our paper. We have uploaded inference code, self‑correction and model‑averaging implementations as supplementary material; the SFT and RL training code is based on open frameworks. Because of its size, we include a subset of the training data in the supplement, and, upon acceptance, we commit to releasing all these to the public.

---

Overall, the discussion led to substantial clarifications and new experiments on formalizer validation, decontamination, additional benchmarks, self‑correction, and model averaging, and addressed all reviewer concerns; we also note that one reviewer explicitly raised their score to 8 and two others indicated they would raise their scores before the rollback

---

### Meta-Review · Area_Chair_ryuG · 2026-01-07

**Summary:**

Here is  a summary of the principal reviewers' concerns :

 - Generalization not convincingly demonstrated beyond MiniF2F/Putnam-style competition domains, need stronger OOD/advanced-domain evaluation (sCYS, 5oHs, nJcq).
 - Reproducibility and transparency gaps.  Incomplete/unclear release (code/data/supplement), missing compute & API costs, training specifics, and hyperparameters (esp. model averaging $\alpha$) (sCYS, BEpZ, 5oHs, nJcq).
 - Training-data provenance and  contamination/decontamination adequacy. Unclear sourcing, possible overlap/leakage via AoPS/benchmark families, and whether checks go beyond exact formal matches (incl. effects of distillation/repetition) (BEpZ, nJcq).
 - Autoformalizer evaluation validity. Small/opaque evaluation set, reliance on LLM judges, risk of “judge hacking”, need clearer independent/human verification and standard benchmarks (BEpZ, 5oHs, nJcq).
 - Attribution and fairness of claimed gains. Unclear ablations (model averaging vs RL), and potentially misleading compute comparisons for revision/self-correction (pass@n vs effective attempts / FLOPs / tokens / wall-clock) (BEpZ, 5oHs, nJcq).

**Reviewer Concerns:**

Here is how these principal concerns were addressed :

 - Generalization beyond MiniF2F/Putnam-style domains. Largely addressed by added ProofNet + RAG, plus FIMO and DeepSeek-ProverBench evaluations, remaining gap is mostly “philosophical” (what counts as reasoning vs library memorization) rather than missing experiments. (sCYS, 5oHs, nJcq)
 - Reproducibility and transparency gaps. Partially addressed. They added dataset sizes for S_1,S_2,S_3, GPU-hour breakdowns, the averaging \alpha, and the Claude API cost, they also uploaded inference and model-averaging code. Still somewhat outstanding because full end-to-end reproducibility depends on post-acceptance release of the full training data and complete training pipeline packaging. (sCYS, BEpZ, 5oHs, nJcq)
 - Training-data provenance and contamination/decontamination adequacy. Somewhat addressed but still the most “fragile” point. Authors clarified data sources, described exact matching on formal statements, and later claimed additional informal string matching with no overlaps, the reviewer ultimately accepted. However, the underlying concern (semantic/fuzzy overlap, benchmark-family leakage, distillation effects) remains a residual risk even after these checks. (BEpZ, nJcq)
 - Autoformalizer evaluation validity. Mostly addressed. They added MiniF2F evaluation, clarified OmniMath, documented the LLM+Lean checking protocol, used a strict consensus judge, and did a small human audit (92% agreement). Remaining gap is scale/independence of human evaluation and potential judge dependence, but it was largely resolved for the reviewers. (BEpZ, 5oHs, nJcq)
 - Attribution and fairness of claimed gains. Partially addressed. They expanded ablations for model averaging ($\alpha$ sweeps, $\alpha=1$ as no-averaging) and revised the revision/self-correction compute comparison using token budgets, longer-context experiments, and wall-clock compilation estimates. Still somewhat outstanding because “fairness” depends on which compute metric one accepts (attempts vs tokens vs FLOPs vs wall-clock) and the gains at low pass@\!k remain modest. (BEpZ, 5oHs, nJcq)

Bottom line. Most concerns were addressed through added evaluations (ProofNet/FIMO/DeepSeek-ProverBench), added compute/cost details, ablations, and supplementary code. The main outstanding risk across reviewers is reproducibility being conditional on post-acceptance release of the full training dataset and, to a lesser extent, fully self-contained training code.

**Reviewer Scores:**

Reviewer nJcq: 6->8 , Reviewer sCYS could also have increased their score 6->8, the remaining two reviewers would probably at least have kept their scores.
Authors are asked to provide access to the full training dataset and to fully self-contained training code with the camera ready submission.

---

### Decision · Program_Chairs · 2026-01-26

Accept (Poster)